# V2P: Visual Attention Calibration for GUI Grounding via Background Suppression and Center Peaking

## Abstract

Precise localization of GUI elements is crucial for the development of GUI agents. Traditional methods rely on bounding box or center-point regression, neglecting spatial interaction uncertainty and visual-semantic hierarchies. Recent methods incorporate attention mechanisms but still face two key issues: (1) ignoring processing background regions causes attention drift from the desired area, and (2) uniform modeling the target UI element fails to distinguish between its center and edges, leading to click imprecision. Inspired by how humans visually process and interact with GUI elements, we propose the Valley-to-Peak (V2P) method to address these issues. To mitigate background distractions, V2P introduces a suppression attention mechanism that minimizes the model's focus on irrelevant regions to highlight the intended region. For the issue of center-edge distinction, V2P applies a Fitts' Law-inspired approach by modeling GUI interactions as 2D Gaussian heatmaps where the weight gradually decreases from the center towards the edges. The weight distribution follows a Gaussian function, with the variance determined by the target's size. Consequently, V2P effectively isolates the target area and teaches the model to concentrate on the most essential point of the UI element. The model trained by V2P achieves the performance with 92.3% and 50.5% on two benchmarks ScreenSpot-v2 and ScreenSpot-Pro. Ablations further confirm each component's contribution, underscoring V2P's generalizability in precise GUI grounding tasks and its potential for real-world deployment in future GUI agents.

## 1 Introduction

Recent advances in large language models (LLMs) and vision-language models (VLMs) have enabled agents to interpret natural language instructions and interact with graphical user interfaces (GUIs) across desktop, mobile, and web platforms. Central to this capability is GUI grounding, which aligns language commands with semantically relevant UI elements and their spatial locations (Cheng et al., 2024). This task bridges user intent and interface actions, supporting the development of intelligent, general-purpose agents for real-world human-computer interaction.

Early approaches framed GUI grounding as coordinate generation task, outputting a bounding box or $(x, y)$ coordinate for a natural-language query (Zhang et al., 2025; Qin et al., 2025). However, this "coordinate generation" method suffers weak spatial–semantic alignment (Wu et al., 2025), treating coordinates like ordinary words without inherent spatial meaning. Moreover, point-wise regression contradicts the multi-point validity inherent in real interactions. Recent work addresses these issues by leveraging the model's attention maps (Wu et al., 2025). Instead of predicting coordinates, it extracts cross-modal attention weights linking instruction tokens to image patches, selecting the most attended patch as the click position. This approach offers dense spatial supervision and naturally tolerates multiple valid click regions, aligning better with human behavior.

However, after manually scrutinizing the attention heatmap of these methods mentioned above (see Sec. 4.3), we found two main issues, as shown in Fig. 1:

1. **Background Distraction**: Current loss functions only reward attention on target patches but fail to penalize it on the background. This leads to a "divergent" attention distribution where background regions also receive high scores. Consequently, softmax normalization

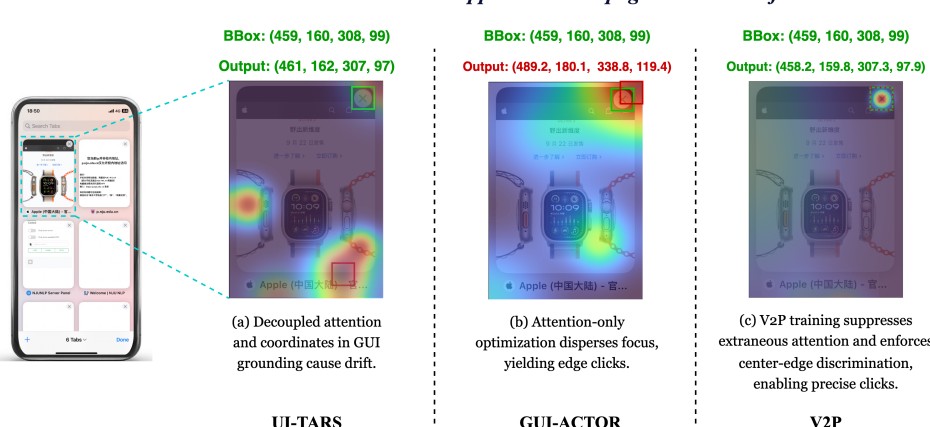

Figure 1: Comparison of different strategies in the GUI grounding task. The green box marks the ground-truth bounding box, and the red box highlights the region where the model places the highest attention given the instruction and screenshot. The overlaid heatmap is colour-coded from cool (blue) to warm (red), with warmer colours indicating higher attention values.

allows these regions to absorb probability mass, weakening or even shifting the intended attention peak.

2. **Centre-edge Confusion**: Because labels treat all pixels within a bounding box equally, the model cannot differentiate an element's center from its edges, resulting in uniform attention and inaccurate clicks that miss the center. Furthermore, for small elements, this often leads the attention to drift towards the edges, making the model more prone to mislocalization, especially when elements overlap.

This raises a key question: *How can we guide the model's attention to focus more precisely on the target UI element?* Motivated by human behavior—first isolating the target (valley suppression) then focusing on the action point (peak emphasis)—we propose **Valley-to-Peak (V2P)**. V2P suppresses distractions by creating low-attention "valleys" in irrelevant areas while sharpening a "peak" at the actionable center.

**Suppression Attention:** We apply inverse attention regularization (Li et al., 2018) to penalize high attention outside the target, isolating true UI elements and reducing attention to non-target regions.

**Fitts-Gaussian Peak Modeling:** Inspired by Fitts' Law (MacKenzie, 1992; Fitts, 1954), we use a 2D Gaussian centered on the target, scaled to its size, to model human's click likelihood, which yields a heatmap that peaks at the center and decays towards the edges, better matching real user interactions.

Together, these modules reshape the attention map, enhancing grounding precision by aligning the model's focus with human patterns.

Our contribution can be summarized as follows:

1. We systematically analyze existing attention-based methods for visual grounding in GUI agents and, through statistical evaluation, identify two main issues——*Background Distraction* and *Center-Edge Confusion*. In addition, we provide a detailed analysis of the underlying causes of these issues and provide insights for further improvements.

2. We introduce *Attention Suppression Mechanism (SA)* to mitigate Background Distraction and employ *Fitts-Gaussian Peak Modeling (FGPM)* to effectively alleviate Center-Edge Confusion. Building on these methods, we propose the **Valley-to-Peak (V2P)** framework, an agentic learning paradigm for GUI grounding that significantly enhances the localization precision and accuracy of Vision-Language Models on GUI elements.

3. Extensive experiments demonstrate that V2P achieves advanced performance on multiple public benchmarks, reaching 92.3% on ScreenSpot-v2 and 50.5% on the challenging ScreenSpot-Pro,

with relative improvements of 3.5% and 23.7%. Furthermore, we confirm that V2P demonstrates significant practical value for real-world deployment and seamless integration into GUI agents.

## 2 RELATED WORK

### 2.1 GUI-AGENTS

GUI agents have progressed from rudimentary random- or rule-based test tools to multimodal, LLM-driven systems that can follow natural-language instructions. Early efforts such as Monkey testing (Wetzlmaier et al., 2016) and planning or script record-and-replay frameworks (Memon et al., 2001; Steven et al., 2000) provided basic coverage but required hand-crafted rules or scripts. Machine-learning techniques later enabled more adaptive behaviour: Humanoid (Li et al., 2020) and Deep GUI (YazdaniBanafsheDaragh & Malek, 2022) learned user-like action policies from screenshots, while widget detectors (White et al., 2019) improved element recognition. Natural-language interfaces soon followed, e.g. FLIN (Mazumder & Riva, 2021) and RUSS (Xu et al., 2021), and reinforcement learning environments like WoB (Shi et al., 2017) and WebShop (Yao et al., 2023) pushed web-scale interaction. The recent arrival of LLMs has unified perception, reasoning and control: WebAgent (Gur et al., 2024) and WebGUM (Furuta et al., 2024) achieve open-world browsing, AutoDroid (Wen et al., 2024) and AppAgent (Zhang et al., 2023) automate smartphones, and desktop agents such as UFO (Zhang et al., 2024) demonstrate GPT-4-level capabilities; industrial systems (e.g. Claude 3.5 Sonnet and Operator) further attest to the practical traction of GUI agents.

### 2.2 GUI GROUNDING

Early works on GUI grounding treated it as a coordinate regression task (Zhang et al., 2025; Qin et al., 2025). However, modern methods have largely shifted to leveraging the cross-modal attention maps of Vision-Language Models (VLMs) (Cheng et al., 2024; Wu et al., 2025). In this paradigm, the model's prediction is derived from the image patch with the highest attention score in response to a language command. While more robust, this approach often suffers from imprecise attention, with focus leaking into irrelevant background regions or spreading too uniformly across the target element. Our work directly addresses this by refining the quality of the attention map itself.

Our approach, V2P, draws inspiration from two distinct areas. To create attention "valleys" and suppress background noise, we adopt attention suppression techniques that penalize focus outside the target region (Li et al., 2018). To form a sharp "peak" at the target's center, we are inspired by both Fitts' Law from Human-Computer Interaction (HCI) (MacKenzie, 1992) and the common practice of using Gaussian heatmaps in localization tasks like pose estimation (Fitts, 1954). To our knowledge, our work is the first to synergistically combine background suppression with center-focused peak modeling to simulate the human pattern of interaction with the UI elements.

## 3 METHOD

We introduce Valley-to-Peak (V2P), a method that reshapes the model's attention landscape to mimic human focus patterns for precise GUI grounding. It achieves this through two synergistic components:

- **Suppression Attention Valley Constraint:** Penalizes attention on irrelevant regions to form low-attention "valleys," effectively suppressing background distractions.

- **Fitts-Gaussian Peak Modeling:** Models interaction likelihood with a size-adaptive 2D Gaussian, creating a sharp attention "peak" at the target's most actionable center.

By jointly optimizing these objectives, V2P produces a continuous, spatially-aware attention map that overcomes the limitations of rigid, uniform labels used in prior work. Below, we first outline the overall architecture (Sec. 3.1), then detail the Suppression Attention (Sec. 3.2) and Fitts-Gaussian Peak Modeling (Sec. 3.3) components.

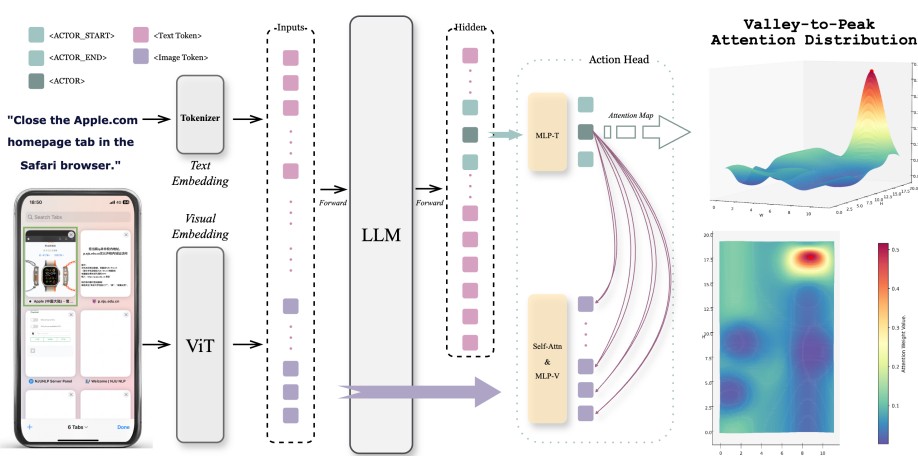

Figure 2: **Valley-to-Peak training method (V2P).** V2P jointly suppresses noise and enhances signals via two strategies: An inverse-attention penalty carves valleys in non-target areas, while size-adaptive Fitts-Gaussian peaks create sharp peaks at UI elements' centers. This dual approach reshapes attention maps (rightmost example), enabling the model to quickly pinpoint interaction points in cluttered interfaces.

## 3.1 MODEL ARCHITECTURE OVERVIEW

We build upon GUI-Actor (Wu et al., 2025), a coordinate-free visual grounding framework that localizes GUI actions through attention rather than coordinate regression. Given a screenshot $I$ and an instruction $q$, the model introduces a special token <ACTOR> in the output sequence as a contextual anchor. The final-layer hidden state of <ACTOR>, denoted $h_{\text{<ACTOR>}}$, is used to compute action attention over image patch features $\{v_1, \ldots, v_M\}$ extracted by the vision encoder.

To enhance spatial coherence among visual patches, we apply a self-attention module over the patch features:

$$\tilde{v}_1, \ldots, \tilde{v}_M = \text{SelfAttn}(v_1, \ldots, v_M), \tag{1}$$

yielding contextualized representations. These are projected into a shared embedding space with $h_{\text{<ACTOR>}}$ via separate MLPs:

$$z = \text{MLP}_T(h_{\text{<ACTOR>}}), \tag{2}$$
$$z_i = \text{MLP}_V(\tilde{v}_i), \quad i = 1, \ldots, M. \tag{3}$$

Attention scores are then computed as:

$$\alpha_i = \frac{z^\top z_i}{\sqrt{d}}, \quad a_i = \frac{\exp(\alpha_i)}{\sum_{j=1}^M \exp(\alpha_j)}, \tag{4}$$

where $d$ is the embedding dimension. The resulting $\{a_i\}_{i=1}^M$ forms a normalized attention distribution over the $M$ image patches, representing the model's belief about the target interaction location.

## 3.2 SUPPRESSION ATTENTION CONSTRAINT FOR DISTRACTION MITIGATION

Attention maps in complex interfaces can suffer from *attention leakage*, where notable responses are mistakenly assigned to regions far from the target area, particularly in the presence of visually similar distracting patches. To address this issue and enhance spatial precision, we propose a Suppression Attention Constraint. This mechanism explicitly penalizes attention allocated to non-target regions, enforcing sparsity and improving the model's ability to distinguish targets from surrounding distractions.

Let $\mathcal{G} \subset \{1, \ldots, M\}$ denote the set of patch indices whose spatial support $R_i$ has empty intersection with the ground-truth bounding box $b$:

$$\mathcal{G} = \{i \in \{1, \ldots, M\} \mid R_i \cap b = \emptyset\}. \tag{5}$$

We define the attention loss as the total attention mass over these irrelevant regions:

$$\mathcal{L}_{\text{Attn}} = \sum_{i \in \mathcal{G}} a_i. \tag{6}$$

To better understand the theoretical foundation of this constraint, we analyze the gradient dynamics of attention weights. For the target patch $k$ with attention weight $A_k = \text{softmax}(s_k)$, the gradient with respect to any non-target patch logit $s_i$ is:

$$w_i = \frac{\partial A_k}{\partial s_i} = \frac{\partial \text{softmax}(s_k)}{\partial s_i} = -\frac{e^{s_k} e^{s_i}}{(\sum_i^M e^{s_i})^2} = -A_k A_i < 0 \quad (i \neq k). \tag{7}$$

This gradient analysis reveals that any increase in attention logits $s_i$ for non-target patches negatively impacts the target attention $A_k$. The magnitude $|w_i| = A_k A_i$ quantifies this negative influence: larger values indicate that even small increases in attention to patch $i$ will cause rapid degradation in target attention $A_k$. This theoretical insight naturally motivates using $|w_i|$ as a weighting factor in our suppression loss, providing stronger penalties for patches that pose greater threats to target attention focus. And we have the *suppression attention loss* combined with gradient weight as:

$$\mathcal{L}_{\text{Sup\_Attn}} = \sum_{i \in \mathcal{G}} w_i a_i. \tag{8}$$

This loss encourages the model to suppress attention on irrelevant regions, thereby reducing the impact of distracting elements in cluttered interfaces. By explicitly minimizing $\mathcal{L}_{\text{Sup\_Attn}}$, the model is incentivized to concentrate its focus on the target region, resulting in enhanced spatial precision and improved robustness.

### 3.3 Fitts-Gaussian Peak Modeling for Center-Focused Grounding

While the Suppression Attention Constraint encourages focus on target regions, overlapping UI elements can still lead to attention dispersion—particularly toward the boundaries of positively labeled components—resulting in ambiguous and spatially diffused attention maps.

Our supervision strategy is inspired by Fitts' Law (MacKenzie, 1992; Fitts, 1954), which reveals that click probability peaks at the center of an UI element and decays toward its edges, closely following a Gaussian distribution. We encode this behavior with Fitts-Gaussian Peak Modeling to guide the model's focus in line with observed human interaction.

Specifically, we model the ideal attention distribution as a 2D Gaussian density centered at the centroid of the ground-truth bounding box $b = [x_1, y_1, x_2, y_2]$:

$$\mu = (c_x, c_y) = \left( \frac{x_1 + x_2}{2}, \frac{y_1 + y_2}{2} \right). \tag{9}$$

To reflect the interaction tolerance associated with target size, we set the standard deviation of the Gaussian proportional to the element's width and height:

$$\sigma_x = \frac{w}{\sigma_{\text{factor}}}, \quad \sigma_y = \frac{h}{\sigma_{\text{factor}}}, \tag{10}$$

where $w = x_2 - x_1$, $h = y_2 - y_1$, and $\sigma_{\text{factor}}$ is a hyperparameter controlling the concentration of the attention prior. This formulation ensures that larger elements—more tolerant to pointing errors—induce broader attention peaks, while smaller elements require sharper focus.

Given an input image partitioned into $M = H \times W$ non-overlapping patches of size $s \times s$, we compute the expected attention mass for each patch $i$, covering spatial region $R_i = [x_{\min}^i, x_{\max}^i] \times [y_{\min}^i, y_{\max}^i]$, by integrating the 2D Gaussian density over $R_i$:

$$y_i = \int_{R_i} \mathcal{N}(x, y; \mu, \Sigma) dx \, dy, \tag{11}$$

where $\Sigma = \text{diag}(\sigma_x^2, \sigma_y^2)$. Thanks to axis-aligned separability, this integral decomposes efficiently into the product of two univariate cumulative distribution functions (CDFs):

$$y_i = \left[ \Phi(x_{\max}^i; c_x, \sigma_x) - \Phi(x_{\min}^i; c_x, \sigma_x) \right] \cdot \left[ \Phi(y_{\max}^i; c_y, \sigma_y) - \Phi(y_{\min}^i; c_y, \sigma_y) \right], \tag{12}$$

with $\Phi(\cdot\,;\mu,\sigma)$ denoting the CDF of a univariate normal distribution.

To supervise the model's predicted attention distribution $\{a_i\}$, we adopt the action attention loss from GUI-Actor (Wu et al., 2025), using the Kullback-Leibler (KL) divergence to measure the discrepancy between the target $p$ and prediction $a$:

$$\mathcal{L}_{\text{Action\_Attn}} = \sum_{i=1}^{M} p_i \log \frac{p_i}{a_i}, \quad p_i = \frac{y_i}{\sum_{j=1}^{M} y_j + \epsilon}, \quad i = 1, \ldots, M, \tag{13}$$

where $\epsilon$ is a small constant for numerical stability.

Fitts-Gaussian Peak Modeling establishes a center-biased, size-aware attention prior that closely mimics human pointing behavior. By discouraging boundary leakage and promoting centralized attention in a graded, interaction-informed manner, it enhances localization precision and improves robustness in complex and cluttered UI layouts.

### 3.4 Valley-to-Peak Training

The overall training objective combines next-token prediction loss with action-focused attention losses:

$$\mathcal{L} = \mathcal{L}_{\text{NTP}} + \lambda_1 \mathcal{L}_{\text{Sup\_Attn}} + \lambda_2 \mathcal{L}_{\text{Action\_Attn}}, \tag{14}$$

where $\mathcal{L}_{\text{Sup\_Attn}}$ suppresses attention outside the target region (Section 3.2), and $\mathcal{L}_{\text{Action\_Attn}}$ enforces alignment between predicted attention and a Gaussian-shaped target distribution (Section 3.3).

Minimizing the combined loss supports a *Valley-to-Peak* training paradigm: coarse suppression followed by fine-grained alignment. $\mathcal{L}_{\text{Sup\_Attn}}$ first suppresses distractions, guiding attention toward the target region. Then, $\mathcal{L}_{\text{Action\_Attn}}$ sharpens this focus by prioritizing the target's center. This reduces misclicks and alleviates ambiguity caused by overlapping labels, ensuring precise and human-like attention alignment. The coarse-to-fine control enables robust interaction predictions, even in dense and visually complex UI environments.

## 4 Experiment

### 4.1 Experiment Setup

**Setup.** We use Qwen2.5-VL-7B-Instruct (Bai et al., 2025) as our backbone and train it on 0.7M filtered GUI screenshots, with a learning rate of 5e-6 and Gaussian factor $\sigma$=1. We evaluate on ScreenSpot-v2 (Wu et al., 2024b) and the more challenging ScreenSpot-Pro (Li et al., 2025) benchmarks using Element Accuracy. Comprehensive implementation details, including the data filtering process, are provided in App. A and B.

### 4.2 Main Result

Our proposed **V2P-7B** demonstrates outstanding performance across diverse benchmarks, showcasing robust generalization and superior efficiency. On the highly challenging ScreenSpot-Pro benchmark, which serves as a strong indicator of out-of-distribution (OOD) generalization, V2P-7B achieves an average accuracy of 50.54% (Tab. 1). This result significantly outperforms all GUI-specific models, including strong RL-based methods like SE-GUI-7B (47.3%) and GUI-G$^2$-7B (47.5%). Remarkably, our 7B model even surpasses the much larger 72B-parameter UI-TARS-72B (38.1%), highlighting exceptional parameter efficiency. This strong performance is consistent across diverse scenarios, with our model securing top scores in 6 of 12 task categories and demonstrating stable adaptability in specialized domains like CAD, Creative, and Science. Furthermore, V2P-7B also excels on the ScreenSpot-v2 benchmark with an average accuracy of 92.3% and we report the result in the Appendix (See Tab. 5).

These advancements are driven by our dual-optimization strategy: *Suppression Attention* mitigates background distractions, while *Fitts-Gaussian Labeling* resolves center-edge confusion. This strong performance is achieved via supervised fine-tuning (SFT) alone, which highlights the potential for

further enhancements through reinforcement learning (RL) integration. The stability of our SFT approach is further evidenced by the model's training trajectory on ScreenSpot-Pro (Fig. 4(c)), which shows no signs of persistent overfitting, unlike baselines that exhibit a continued performance decline. The consistent gains across diverse UI platforms and interaction types affirm V2P's robust generalizability for real-world GUI grounding applications.

| Model | ScreenSpot-Pro Accuracy (%) | | | | | | | | | | | | | | |
| | CAD | | Dev | | Creative | | Scientific | | Office | | OS | | Avg. | | |
| | Text | Icon | Text | Icon | Text | Icon | Text | Icon | Text | Icon | Text | Icon | Text | Icon | **Avg.** |
|---|---|---|---|---|---|---|---|---|---|---|---|---|---|---|---|
| *Proprietary Models* | | | | | | | | | | | | | | | |
| GPT-4o | 2.0 | 0.0 | 1.3 | 0.0 | 1.0 | 0.0 | 2.1 | 0.0 | 1.1 | 0.0 | 0.0 | 0.0 | 1.3 | 0.0 | 0.8 |
| Claude Computer Use | 14.5 | 3.7 | 22.0 | 3.9 | 25.9 | 3.4 | 33.9 | 15.8 | 30.1 | 16.3 | 11.0 | 4.5 | 23.4 | 7.1 | 17.1 |
| *General Open-source Models* | | | | | | | | | | | | | | | |
| Qwen2.5-VL-3B | 9.1 | 7.3 | 22.1 | 1.4 | 26.8 | 2.1 | 38.2 | 7.3 | 33.9 | 15.1 | 10.3 | 1.1 | 23.6 | 3.8 | 16.1 |
| Qwen2.5-VL-7B | 16.8 | 1.6 | 46.8 | 4.1 | 35.9 | 7.7 | 49.3 | 7.3 | 52.5 | 20.8 | 37.4 | 6.7 | 38.9 | 7.1 | 26.8 |
| *GUI-specific Models (SFT)* | | | | | | | | | | | | | | | |
| SeeClick-9.6B | 2.5 | 0.0 | 0.6 | 0.0 | 1.0 | 0.0 | 3.5 | 0.0 | 1.1 | 0.0 | 2.8 | 0.0 | 1.8 | 0.0 | 1.1 |
| FOCUS-2B | 7.6 | 3.1 | 22.8 | 1.7 | 23.7 | 1.7 | 25.0 | 7.1 | 23.2 | 7.7 | 17.8 | 2.5 | 19.8 | 3.9 | 13.3 |
| CogAgent-18B | 7.1 | 3.1 | 14.9 | 0.7 | 9.6 | 0.0 | 22.2 | 1.8 | 13.0 | 0.0 | 5.6 | 0.0 | 12.0 | 0.8 | 7.7 |
| Aria-UI | 7.6 | 1.6 | 16.2 | 0.0 | 23.7 | 2.1 | 27.1 | 6.4 | 20.3 | 1.9 | 4.7 | 0.0 | 17.1 | 2.0 | 11.3 |
| OS-Atlas-7B | 12.2 | 4.7 | 33.1 | 1.4 | 28.8 | 2.8 | 37.5 | 7.3 | 33.9 | 5.7 | 27.1 | 4.5 | 28.1 | 4.0 | 18.9 |
| ShowUI-2B | 2.5 | 0.0 | 16.9 | 1.4 | 9.1 | 0.0 | 13.2 | 7.3 | 15.3 | 7.5 | 10.3 | 2.2 | 10.8 | 2.6 | 7.7 |
| UGround-7B | 14.2 | 1.6 | 26.6 | 2.1 | 27.3 | 2.8 | 31.9 | 2.7 | 31.6 | 11.3 | 17.8 | 0.0 | 25.0 | 2.8 | 16.5 |
| UGround-V1-7B | 15.8 | 1.2 | 51.9 | 2.8 | 47.5 | 9.7 | 57.6 | 14.5 | 60.5 | 13.2 | 38.3 | 7.9 | 45.2 | 8.1 | 31.1 |
| UI-TARS-2B | 17.8 | 4.7 | 47.4 | 4.1 | 42.9 | 6.3 | 56.9 | 17.3 | 50.3 | 17.0 | 21.5 | 5.6 | 39.6 | 8.4 | 27.7 |
| UI-TARS-7B | 20.8 | 9.4 | 58.4 | 12.4 | 50.0 | 9.1 | 63.9 | 31.8 | 63.3 | 20.8 | 30.8 | 16.9 | 47.8 | 16.2 | 35.7 |
| UI-TARS-72B | 18.8 | 12.5 | 62.9 | 17.2 | 57.1 | 15.4 | 64.6 | 20.9 | 63.3 | 26.4 | 42.1 | 15.7 | 50.9 | 17.6 | 38.1 |
| JEDI-3B | 27.4 | 9.4 | 61.0 | 13.8 | 53.5 | 8.4 | 54.2 | 18.2 | 64.4 | 32.1 | 38.3 | 9.0 | 49.8 | 13.7 | 36.1 |
| JEDI-7B | 38.0 | 14.1 | 42.9 | 11.0 | 50.0 | 11.9 | 72.9 | 25.5 | 75.1 | 47.2 | 33.6 | 16.9 | 52.6 | 18.2 | 39.5 |
| GUI-Actor-7B | – | – | – | – | – | – | – | – | – | – | – | – | – | – | 44.6 |
| *GUI-specific Models (RL)* | | | | | | | | | | | | | | | |
| UI-R1-3B | 11.2 | 6.3 | 22.7 | 4.1 | 27.3 | 3.5 | 42.4 | 11.8 | 32.2 | 11.3 | 13.1 | 4.5 | 24.9 | 6.4 | 17.8 |
| UI-R1-E-3B | 37.1 | 12.5 | 46.1 | 6.9 | 41.9 | 4.2 | 56.9 | 21.8 | 65.0 | 26.4 | 32.7 | 10.1 | – | – | 33.5 |
| GUI-R1-3B | 26.4 | 7.8 | 33.8 | 4.8 | 40.9 | 5.6 | 61.8 | 17.3 | 53.6 | 17.0 | 28.1 | 5.6 | – | – | – |
| GUI-R1-7B | 23.9 | 6.3 | 49.4 | 4.8 | 38.9 | 8.4 | 55.6 | 11.8 | 58.7 | 26.4 | 42.1 | 16.9 | – | – | – |
| InfiGUI-R1-3B | 33.0 | 14.1 | 51.3 | 12.4 | 44.9 | 7.0 | 58.3 | 20.0 | 65.5 | 28.3 | 43.9 | 12.4 | 49.1 | 14.1 | 35.7 |
| GUI-G1-3B | 39.6 | 9.4 | 50.7 | 10.3 | 36.6 | 11.9 | 61.8 | 30.0 | 67.2 | 32.1 | 23.5 | 10.6 | 49.5 | 16.8 | 37.1 |
| SE-GUI-3B | 38.1 | 12.5 | 55.8 | 7.6 | 47.0 | 4.9 | 61.8 | 16.4 | 59.9 | 24.5 | 40.2 | 12.4 | 50.4 | 11.8 | 35.9 |
| SE-GUI-7B | 51.3 | 42.2 | 68.2 | 19.3 | 57.6 | 9.1 | 75.0 | 28.2 | 78.5 | 43.4 | 49.5 | 25.8 | 63.5 | 21.0 | 47.3 |
| GUI-G$^2$-7B | 55.8 | 12.5 | 68.8 | 17.2 | 57.1 | 15.4 | 77.1 | 24.5 | 74.0 | 32.7 | 57.9 | 21.3 | 64.7 | 19.6 | 47.5 |
| *Ours* | | | | | | | | | | | | | | | |
| **V2P-7B** | **58.38** | 12.50 | 67.53 | **24.83** | 62.63 | 16.08 | 73.61 | **33.64** | 75.71 | 43.40 | 56.07 | **32.58** | 65.81 | 25.83 | **50.54** |

Table 1: Comparison of Model Performance Across Task Categories in ScreenSpot-Pro. Bold text highlights the best results, while "–" represents missing values not reported in the original papers. The baseline models utilize various backbones and parameter sizes, as indicated by their names (e.g., -7B, -18B). Further details are provided in App. C.

## 4.3 ATTENTION MAP QUALITY ANALYSIS

To diagnose common failure modes in GUI grounding, we manually analyzed the attention quality of 100 randomly sampled cases across our V2P model and two representative baselines (UI-TARS (Qin et al., 2025) and GUI-Actor (Wu et al., 2025)). Our analysis focused on two critical issues: **background distraction** (attention on irrelevant regions) and **center-edge confusion** (imprecise localization at the element's boundary).

The results, summarized in Table 2, reveal a clear discrepancy in attention quality. Traditional textual-output models like UI-TARS suffer from near-total background distraction (100 cases), indicating that coordinate supervision fails to teach visual focus. While vision-attention models like GUI-Actor show improvement (74 total issues), they still struggle with background distraction and center-edge confusion. In contrast, our V2P model demonstrates superior performance, reducing background distraction to only 42 cases and center-edge confusion to 15. With a total of just 57 issues, V2P significantly outperforms both baselines, providing direct evidence that its explicit design effectively remedies these common failure modes for more reliable GUI grounding.

| Attention Issue | UI-TARS | GUI-Actor | **V2P (Ours)** |
|---|---|---|---|
| Background distraction | 100 | 53 | **42** |
| Centre-edge confusion | 0 | 21 | **15** |
| **Total Issues** | 100 | 74 | **57** |

Table 2: Attention Map Quality Analysis on 100 Manually Sampled Cases.

## 4.4 ABLATION STUDY

### 4.4.1 ABLATION STUDY FOR V2P

Our ablation study (Tab. 3a) on the challenging ScreenSpot-Pro benchmark validates the efficacy of our V2P method. Removing *Fitts-Gaussian Peak Modeling* and *Suppression Attention* individually causes performance drops of 3.0% (to 47.5%) and 3.2% respectively, highlighting their roles in resolving center-edge confusion and reducing background distractions.

On the simpler ScreenSpot-v2, removing *Fitts-Gaussian Peak Modeling* alone has a negligible impact (92.3% accuracy), as its simple layouts with minimal overlap diminish the need for precise center-point guidance. We further demonstrate in Sec. 4.4.2 that this component excels on complex, overlapping interfaces. However, removing both components still results in a slight drop to 91.9%. This shows that while V2P's full potential is most evident in challenging scenarios like ScreenSpot-Pro, it remains robust across different complexities.

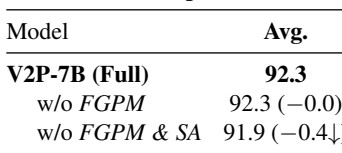

| ScreenSpot-Pro | |
|---|---|
| Model | **Avg.** |
| **V2P-7B (Full)** | **50.5** |
| w/o *FGPM* | 47.5 (−3.0↓) |
| w/o *FGPM & SA* | 44.3 (−6.2↓) |
| ScreenSpot-v2 | |
| Model | **Avg.** |
| **V2P-7B (Full)** | **92.3** |
| w/o *FGPM* | 92.3 (−0.0) |
| w/o *FGPM & SA* | 91.9 (−0.4↓) |

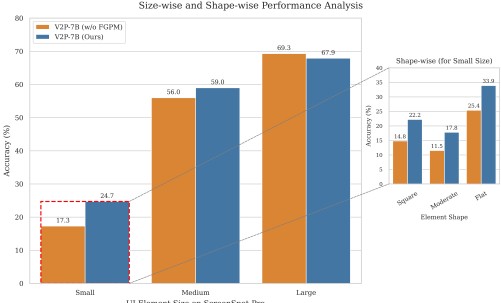

(a) Ablation study on *ScreenSpot-Pro* and *ScreenSpot-v2*.

(b) Ablation study demonstrating the effectiveness of **Fitts-Gaussian Peak Modeling**.

Figure 3: Combined ablation studies. (a) Performance on different datasets. (b) Detailed breakdown for UI element size and shape.

### 4.4.2 ABLATION STUDY FOR EFFECTIVENESS OF FITTS-GAUSSIAN PEAK MODELING

Traditional attention methods often yield overly broad regions, misaligning with small UI elements and producing points outside their boxes (Fig. 1(b)). Fitts-Gaussian Peak Modeling counters this by centering the attention, boosting accuracy on tiny elements. We conduct ablation studies on the challenging ScreenSpot-Pro dataset (Li et al., 2025) to validate our approach.

We first split UI elements into **small**, **medium**, and **large** categories based on bounding box sizes. Fig. 3b shows that our Fitts-Gaussian Peak Modeling (FGPM) yields substantial improvements on challenging smaller elements: 7.4% for **small** and 3.0% for **medium** elements. For **large** elements, there is a slight decrease, as original attention-based methods with dispersed attention may accidentally fall within large bounding boxes even when localizing incorrectly, while our precise targeting reduces such coincidental hits.

We further analyze shape impact by categorizing **small** elements into **square**, **moderate**, and **flat** shapes based on aspect ratios. The zoomed-in table in Fig. 3b demonstrates consistent improve-

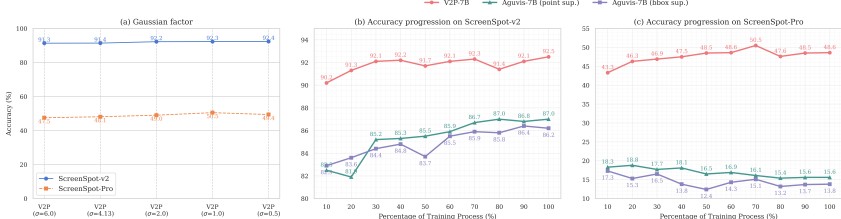

Figure 4: Ablation Study of Fitts-Gaussian Peak Modeling and Generalization Analysis of V2P. The table shows the performance impact of our proposed method and its generalization capability on an out-of-distribution dataset. Results for Aguvis-7B are from GUI-Actor (Wu et al., 2025).

ments across all shapes, confirming that FGPM effectively addresses precise localization challenges regardless of element shape.

### 4.4.3 ABLATION STUDY FOR GAUSSIAN FACTOR $\sigma$

We conducted ablation experiments to analyze the effect of different Gaussian factors $\sigma$ on model performance. As shown in Fig. 4(a), the model's performance is strongly influenced by the choice of Gaussian factor $\sigma$. For both ScreenSpot-v2 and ScreenSpot-Pro, accuracy improves as $\sigma$ decreases. For example, on ScreenSpot-v2, the accuracy rises from 91.3% at $\sigma = 6.0$ to 92.4% at $\sigma = 0.5$, while ScreenSpot-Pro achieves its best result of 50.5% accuracy at $\sigma = 1.0$.

We suspect that this is because that larger $\sigma$ values correspond to a broader Gaussian distribution, which tends to dilute the spatial focus and introduce noise into the attention maps. In comparison, smaller $\sigma$ produces sharper Gaussian peaks, allowing the model to localize UI elements with higher precision and resulting in more accurate click predictions. These results underscore that carefully balancing the Gaussian factor is crucial: excessively large values hinder localization, while moderate to smaller values significantly enhance spatial accuracy and overall model performance.

### 4.5 QUALITATIVE AND ADVANCED CAPABILITIES ANALYSIS

To provide deeper insights beyond quantitative metrics, we conducted a series of qualitative and advanced capability analyses, with full details and visualizations provided in App. D.

Our qualitative review (Fig. 5 and 6) confirms that V2P generates sharp, well-defined attention maps that align closely with target element boundaries, successfully mitigating common failure modes like semantic confusion and low-confidence predictions. Furthermore, we validated V2P's practical utility in more complex scenarios. As shown in Fig. 7(a) and 8, the model demonstrates robust performance in **multi-step interaction** workflows, maintaining contextual awareness across sequential operations. It also exhibits sophisticated **multi-target localization** capabilities in Fig. 7(b), simultaneously identifying multiple elements within a single interface.

Finally, we integrated V2P into an end-to-end agent to tackle a real-world, multi-app task. The model successfully completed the entire 7-step trajectory without error (Fig. 10), confirming its potential as a reliable grounding component for practical GUI automation.

## 5 CONCLUSION

We presented V2P, a novel framework for GUI grounding that operationalizes a "valley-to-peak" strategy. By first suppressing background distractions and then highlighting clickable regions with a Fitts-Gaussian peak, V2P explicitly addresses the critical issues of background distraction and center-edge confusion. Our approach achieves state-of-the-art performance, attaining 92.3% accuracy on ScreenSpot-v2 and 50.5% on the challenging ScreenSpot-Pro benchmark. Extensive experiments validate the effectiveness of each component and demonstrate the strong generalization capabilities of our model. As a lightweight, interpretable, and scalable solution, V2P offers tangible benefits for developing robust GUI agents capable of operating in complex, real-world software environments.

## REPRODUCIBILITY STATEMENT

We have made every effort to ensure the reproducibility of our results. Key experimental settings, datasets, model architectures, and training procedures are described in detail in the main paper (Sec. 4.1 and App. A). The anonymous source code is available via a link in the supplementary materials. We encourage readers to consult these resources for detailed replication guidance.

## ACKNOWLEDGEMENT

During the preparation of this manuscript, we used Google Gemini-2.5-Pro (gem, 2025) to assist with language polishing and proofreading to improve the clarity and readability of the text. The authors assume full responsibility for the final content.

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

# A  TRAINING AND INFERENCE DETAILS

## A.1  SOURCE TRAINING DATA

Following GUI-Actor (Wu et al., 2025), we compile our training dataset from several publicly available, high-quality GUI datasets, with summary statistics provided in Table 3. To ensure fair evaluation, we also exclude any samples from Wave-UI that overlap with the test sets of downstream tasks. Our data recipe is built from several public GUI datasets, the source data totaling approximately 1M screenshots. To ensure annotation quality, we apply Ominiparser (Wan et al., 2024) to detect bounding boxes for all samples and filter those where the IoU between ground truth (GT) and parser-detected boxes is less than 0.3, as such cases likely contain annotation errors, this step improves the data consistency for training. After filtering, there are about $\sim 0.7$M screenshots remains.

| Dataset | # of Elements | # of Screenshots | Platform |
|---|---|---|---|
| Uground Web–Hybrid (Gou et al., 2025a) | 8M | 775K | Web |
| GUI-Env (Chen et al., 2025) | 262K | 70K | Web |
| GUI-Act (Chen et al., 2025) | 42K | 13K | Web |
| AndroidControl (Li et al., 2024) | 47K | 47K | Android |
| AMEX (Chai et al., 2025) | 1.2M | 100K | Android |
| Wave-UI | 50K | 7K | Hybrid |
| **Total** | 9.6M | 1M | – |

Table 3: Overview of training datasets used for GUI-Actor.

## A.2  TRAINING AND INFERENCE SETUP

During the training phase, we first freeze the backbone VLM parameters and train only the action head ($\sim 20$M parameters). In the second phase, we fine-tune the entire model using the filtered dataset with standard supervised learning. At inference, we follow deterministic generation with a temperature of 0 and adopt a confidence threshold of $\gamma = 0.95$ for the ScreenSpot-Pro benchmark and $\gamma = 0.8$ for ScreenSpot-v2 tasks.

## A.3  TRAINING AND INFERENCE COSTS

### A.3.1  TRAINING COSTS

The V2P-7B model was trained using a comprehensive dataset comprising Uground Web-Hybrid, GUI-Env, GUI-Act, AndroidControl, AMEX, and Wave-UI (Tab. 3). The training process was conducted on a high-performance computing cluster equipped with 32 NVIDIA H200 GPUs.

Following the (Wu et al., 2025), the training procedure consisted of two phases: an initial warmup phase which only fine-tunes the parameters of attention heads requiring approximately 15 hours, followed by full supervised fine-tuning (SFT) that took an additional 20 hours. This results in a total training time of approximately 35 hours across 32 H200 GPUs, equivalent to 1,120 GPU-hours for the complete training pipeline.

### A.3.2  INFERENCE EFFICIENCY

Tab. 4 presents the detailed inference performance metrics of the V2P-7B model evaluated on different benchmarks with batch size 1.

| Metric | ScreenSpot-v2 | ScreenSpot-Pro |
|---|---|---|
| Latency per Sample (ms) | 154.00 | 1857.97 |
| Throughput (samples/sec) | 6.49 | 0.54 |

Table 4: Inference Performance Metrics

The model demonstrates efficient inference performance, with significantly faster processing on ScreenSpot-v2 compared to ScreenSpot-Pro, likely due to the complexity differences between the two benchmarks.

### A.3.3 RESOURCE REQUIREMENTS

The V2P-7B model comprises approximately 7 billion parameters and requires around 110 GB of disk storage for the model weights. During inference, the model consumes approximately 80 GB of GPU memory at peak usage. All inference evaluations were conducted on a single NVIDIA H200 GPU with 141 GB of memory, demonstrating that the model can be efficiently deployed on high-end consumer or enterprise-grade hardware.

The computational requirements make the model accessible for research and production environments with sufficient GPU resources, while the inference speeds are suitable for real-time applications, particularly on the ScreenSpot-v2 benchmark.

## B BENCHMARKS

Our evaluation centers on two sophisticated benchmarks for GUI visual grounding: ScreenSpot-v2 (Wu et al., 2024b) and ScreenSpot-Pro (Li et al., 2025).

**ScreenSpot-v2** encompasses 1,272 carefully annotated instructions, each paired with corresponding target elements across diverse GUI environments, including mobile (Android and iOS), desktop (macOS and Windows), and web platforms. The dataset is designed to improve the quality and reliability of GUI visual grounding tasks, addressing key challenges such as eliminating ambiguities in natural language instructions and resolving annotation errors. By refining the alignment between textual descriptions and interface elements, ScreenSpot-v2 provides a robust and standardized benchmark for evaluating grounding models.

**ScreenSpot-Pro**, meanwhile, focuses on more demanding scenarios, especially those involving high-resolution professional applications. It contains 1,581 tasks annotated by domain experts across 23 specialized software applications, spanning three operating systems. This benchmark significantly broadens the scope of GUI visual grounding by introducing interfaces with industrial software and multi-window layouts, creating a larger domain gap compared to most pretraining data. With its increased complexity and domain diversity, ScreenSpot-Pro is an invaluable resource for assessing the generalization ability of models in realistic and challenging GUI environments.

## C BASELINES

### C.1 BASELINES FOR SCREENSPOT-PRO

We establish comprehensive benchmarking across four categories of state-of-the-art GUI understanding systems:

- **Proprietary Systems**: GPT-4o (OpenAI, 2024) (vision-language foundation model), Claude Computer Use (Google, 2024) (specialized GUI agent)
- **General-Purpose Open-Source**: Qwen2.5-VL series (Bai et al., 2025) (7B/72B parameter variants)
- **GUI-Specialized (SFT)**:
  - Medium-scale: SeeClick-9.6B (Based on Qwen-VL-Chat) (Cheng et al., 2024), FOCUS-2B (Based on Qwen2-VL-2B-Instruct) (Zhang et al., 2024), OS-Atlas-7B (Based on Qwen2-VL-7B) (Wu et al., 2024a)
  - Large-scale: CogAgent-18B (Based on CogVLM17B) (Hong et al., 2024), Aria-UI (Based on Megatron-LM) (Yang et al., 2025b), JEDI series (Based on Qwen2.5-VL series) (Xie et al., 2025)
  - Domain-specific: ShowUI-2B (Based on Qwen2-VL-2B) (Lin et al., 2024), Uground series (Based on Qwen2-VL series) (Gou et al., 2025b), UI-TARS series (Based on Qwen2-VL series) (Qin et al., 2025)

| Model | ScreenSpot-v2 Accuracy (%) | | | | | | |
|---|---|---|---|---|---|---|---|
| | Mobile-Text | Mobile-Icon | Desktop-Text | Desktop-Icon | Web-Text | Web-Icon | Avg. |
| *Proprietary Models* | | | | | | | |
| Operator | 47.3 | 41.5 | 90.2 | 80.3 | 92.8 | 84.3 | 70.5 |
| GPT-4o + OmniParser-v2 | 95.5 | 74.6 | 92.3 | 60.9 | 88.0 | 59.6 | 80.7 |
| *General Open-source Models* | | | | | | | |
| Qwen2.5-VL-3B | 93.4 | 73.5 | 88.1 | 58.6 | 88.0 | 71.4 | 80.9 |
| Qwen2.5-VL-7B | 97.6 | 87.2 | 90.2 | 74.2 | 93.2 | 81.3 | 88.8 |
| *GUI-specific Models (SFT)* | | | | | | | |
| SeeClick-9.6B | 78.4 | 50.7 | 70.1 | 29.3 | 55.2 | 32.5 | 55.1 |
| Magma-8B | 62.8 | 53.4 | 80.0 | 57.9 | 67.5 | 47.3 | 61.5 |
| OS-Atlas-4B | 87.2 | 59.7 | 72.7 | 46.4 | 85.9 | 63.1 | 71.9 |
| UI-TARS-2B | 95.2 | 79.1 | 90.7 | 68.6 | 87.2 | 78.3 | 84.7 |
| OS-Atlas-7B | 95.2 | 75.8 | 90.7 | 63.6 | 90.6 | 77.3 | 84.1 |
| Aguvis-7B | 95.5 | 77.3 | 95.4 | 77.9 | 91.0 | 72.4 | 86.0 |
| UGround-V1-7B | 95.0 | 83.3 | 95.0 | 77.8 | 92.1 | 77.2 | 87.6 |
| UI-TARS-72B | 94.8 | 86.3 | 91.2 | 87.9 | 91.5 | 87.7 | 90.3 |
| GUI-Actor-3B | 97.6 | 83.4 | 96.9 | 83.6 | 94.0 | 85.7 | 91.0 |
| UI-TARS-7B | 96.9 | 89.1 | 95.4 | 85.0 | 93.6 | 85.2 | 91.6 |
| GUI-Actor-7B | 97.6 | 88.2 | 96.9 | 85.7 | 93.2 | 86.7 | 92.1 |
| *GUI-specific Models (RL)* | | | | | | | |
| SE-GUI-7B | - | - | - | - | - | - | 90.3 |
| LPO-8B | - | - | - | - | - | - | 90.5 |
| *Ours* | | | | | | | |
| **V2P-7B** | **98.1** | 88.0 | 96.1 | **89.7** | 95.4 | 84.4 | **92.3** |

Table 5: Comparison of Model Performance Across Task Categories in ScreenSpot-v2. Bold text highlights the best results, while "–" represents missing values not reported in the original papers.

- **GUI-Specialized (RL)**:
  - R1-style: UI-R1 (Based on Qwen2.5-VL-3B-Instruct) (Lu et al., 2025), GUI-R1 (Based on Qwen2.5-VL-3B and Qwen2.5-VL-7B) (Luo et al., 2025), InfiGUI-R1-3B (Based on Qwen2.5-VL-3B-Instruct) (Liu et al., 2025)
  - Gaussian-based: GUI-G1-3B (Based on Qwen2.5-VL-3B-Instruct) (Zhou et al., 2025), SE-GUI (Based on Qwen2.5-VL-3B and Qwen2.5-VL-7B) (Yuan et al., 2025), GUI-G$^2$-7B (Based on Qwen2.5-VL-7B-Instruct) (Tang et al., 2025a)

## C.2 BASELINES FOR SCREENSPOT-V2

We establish comprehensive benchmarking across four categories of state-of-the-art GUI understanding systems:

- **Proprietary Systems**: Operator (OpenAI, 2023) (proprietary multimodal system), GPT-4o + OmniParser-v2 (OpenAI, 2024; Wan et al., 2024) (enhanced vision-language integration)
- **General-Purpose Open-Source**: Qwen2.5-VL series (Bai et al., 2025) (7B/72B parameter variants)
- **GUI-Specialized (SFT)**:
  - Medium-scale: SeeClick-9.6B (Cheng et al., 2024), Magma-8B (Yang et al., 2025a), OS-Atlas series (Wu et al., 2024a)
  - Large-scale: UI-TARS series (Qin et al., 2025), Uground series (Gou et al., 2025b), GUI-Actor series (Wu et al., 2025)
  - Domain-specific: Aguvis-7B (Xu et al., 2025)

- **GUI-Specialized (RL)**:
    - SE-GUI-based: SE-GUI-7B (Yuan et al., 2025), LPO-8B (Tang et al., 2025b)

## D    ADDITIONAL EXPERIMENTAL RESULTS AND ANALYSIS

### D.1    QUALITATIVE ANALYSIS OF MODEL PERFORMANCE

#### D.1.1    SUCCESS CASES

Fig. 5 demonstrate several representative success cases where our V2P-7B model achieves accurate GUI element localization. Through these successful examples, we observe that the model exhibits high confidence in precisely highlighting target regions, with attention distributions that closely align with the actual shapes of UI elements. The attention maps show sharp, well-defined boundaries that accurately correspond to button edges, text field borders, and icon contours. This demonstrates the model's robust understanding of visual-semantic correspondence between natural language instructions and GUI components, effectively bridging the gap between textual descriptions and visual interface elements.

#### D.1.2    FAILURE CASES AND ERROR ANALYSIS

Our analysis of failure cases reveals several interesting patterns and limitations, as illustrated in Fig. 6. In some instances, we observe that the model encounters difficulties when multiple UI elements share semantic similarities. The model often exhibits high confidence while incorrectly selecting semantically related but functionally different elements or misidentifying similar icons with different purposes (Fig. 6a).

Additionally, we identify cases where the model's attention distribution becomes highly dispersed across the interface, which we interpret as an indicator of *low confidence* (Fig. 6b). This scattered attention pattern typically occurs in scenarios with numerous distracting elements or cluttered interfaces, suggesting that the model's decision-making process becomes uncertain when faced with complex visual layouts.

Furthermore, we observe failure modes where the model's attention concentrates entirely on regions completely unrelated to the target element (Fig. 6c). These cases often involve ambiguous natural language descriptions or interfaces with unconventional design patterns that deviate from the model's training distribution. Such failures highlight the need for enhanced user intent understanding and more comprehensive UI context comprehension capabilities.

#### D.1.3    MULTI-STEP INTERACTION SCENARIOS

To evaluate the model's capability in complex interaction workflows, we designed multi-step interaction scenarios using pure grounding tasks selected from the AndroidControl (Li et al., 2024) dataset. Fig. 7a and Fig. 8 showcases the model's performance across sequential GUI operations.

The results demonstrate that our model maintains consistent accuracy throughout extended interaction sequences, successfully completing multi-step tasks that require contextual understanding and state awareness. This capability highlights the model's potential for integration into automated GUI Agent frameworks, where reliable multi-step interaction is crucial for practical deployment. We conducted an experiment that incorporate our V2P-7B into an end-to-end real-world application case, more details can be seen in Appendix D.2

#### D.1.4    MULTI-TARGET LOCALIZATION CAPABILITIES

We investigated the model's ability to simultaneously localize multiple targets within a single interface, which holds significant value for batch operations and improving inference efficiency. Fig. 7b presents our experimental setup using a calculator interface, where we tasked the model with simultaneously localizing the elements "1", "0", and "00".

The results reveal that the model successfully generates attention distributions for all three target elements simultaneously, with appropriately differentiated confidence levels. Notably, the element "1"

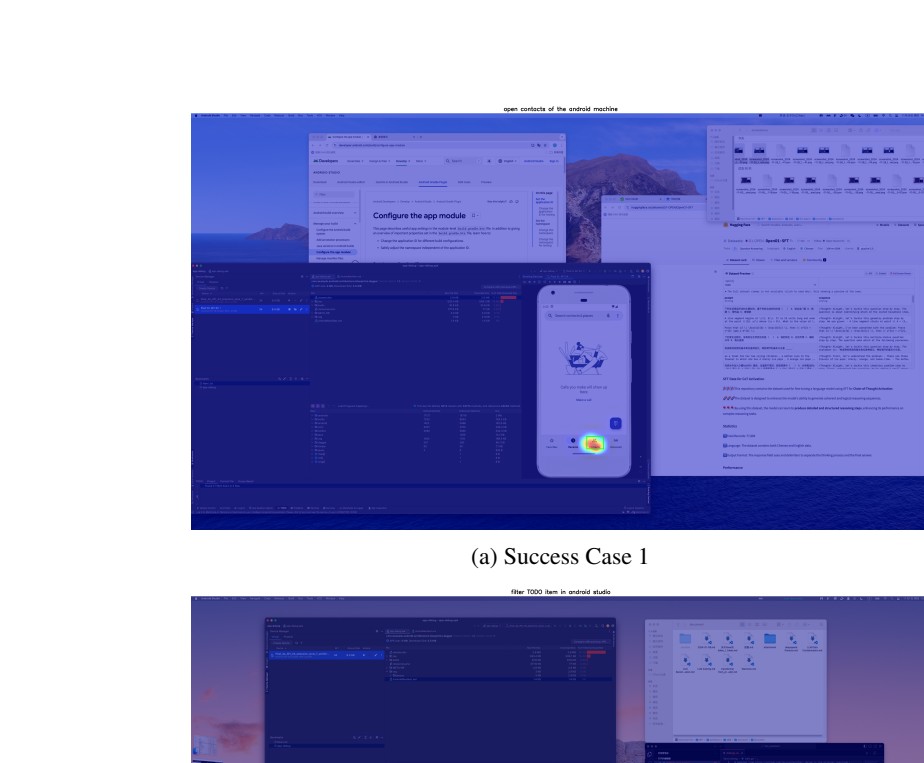

(a) Success Case 1

(b) Success Case 2

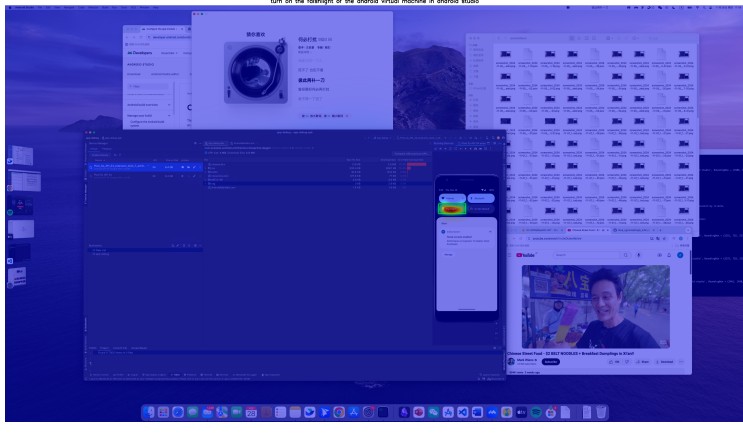

(c) Success Case 3

Figure 5: Representative success cases of GUI element localization.

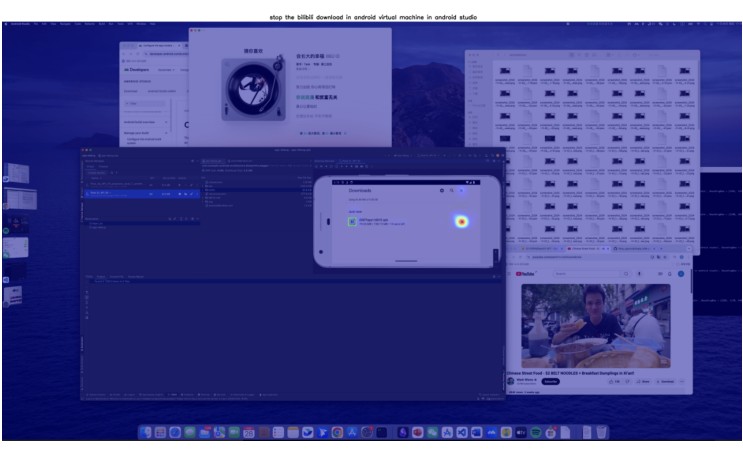

(a) Failure Case 1

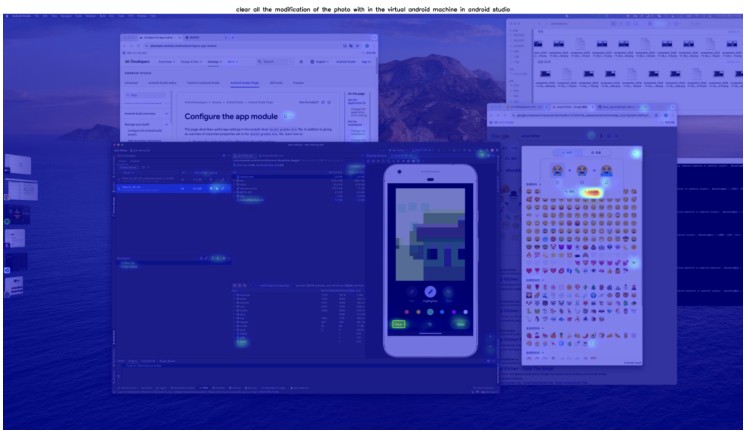

(b) Failure Case 2

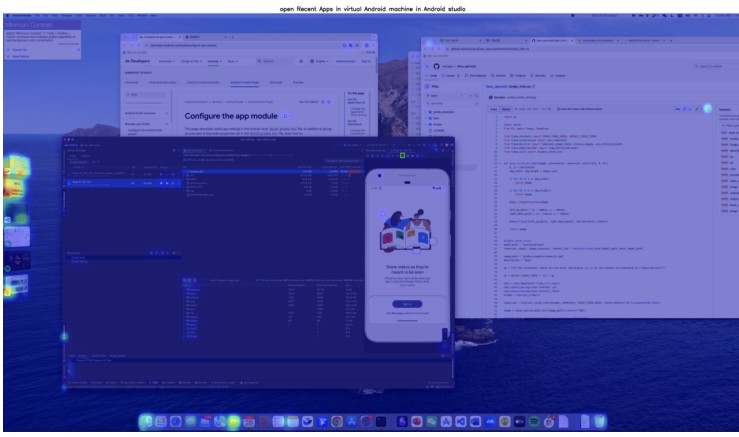

(c) Failure Case 3

Figure 6: Representative failure cases of GUI element localization.

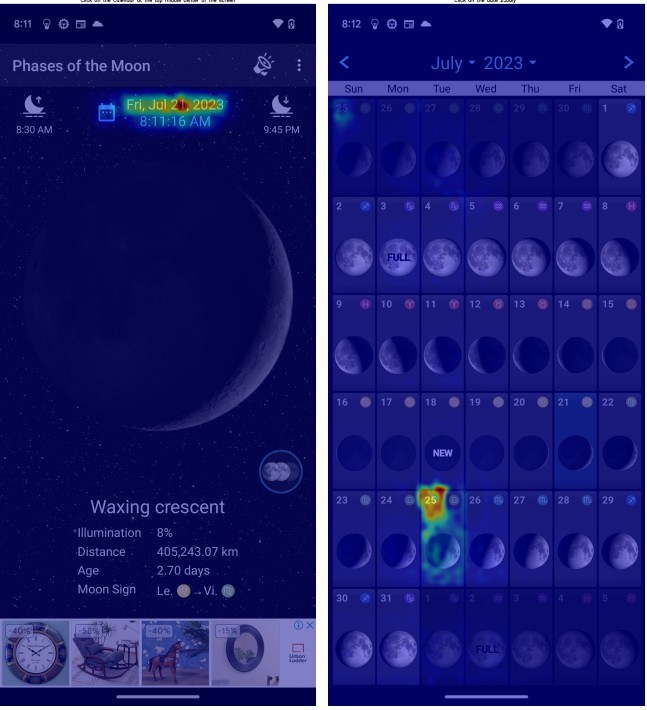
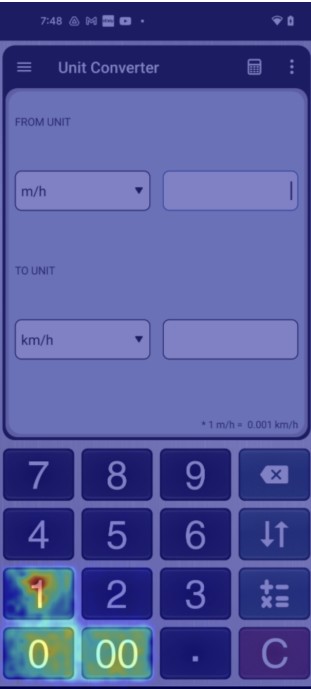

(a) Multi step grounding case 1: "Open Phase of the moon App, select the date 25 July on the calendar and view the moon phase for that date." Step 1 (left) and Step 2 (right).

(b) Multi-target grounding case.

Figure 7: Multi-step grounding case and multi-target grounding case.

receives the highest attention intensity, followed by "0" and "00" respectively, which aligns with the natural priority and visual prominence of these elements. This multi-target capability demonstrates the model's sophisticated attention mechanism and its potential for complex GUI analysis tasks requiring simultaneous element identification, as well as its genuine understanding capability of user queries.

## D.2 END-TO-END REAL-WORLD APPLICATION

### D.2.1 TASK DESIGN AND SETUP

To validate the practical value of our grounding model in real-world scenarios, we selected a complex, complete GUI navigation task from the AndroidControl dataset (Li et al., 2024). The chosen task encompasses a comprehensive action space including *"navigate_back"*, *"click"*, and *"type_text"* operations, spanning a total of 7 sequential steps that effectively simulate realistic user interaction circumstance.

The task involves navigating through a complex multi-application workflow on mobile devices, requiring seamless transitions between different apps (Daff Moon app and Gmail app), information extraction and processing, and email composition with specific recipient details. This scenario was specifically chosen to test the model's ability to handle real-world user requests that span multiple applications, maintain context across app switches, and accurately interpret nuanced user intents involving personal relationships and specific communication requirements. We integrated our V2P-7B model into the grounding components of the navigation pipeline, replacing the baseline grounding module while maintaining the overall task execution framework.

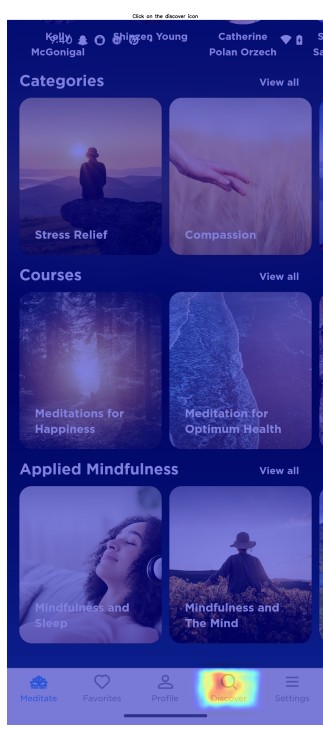

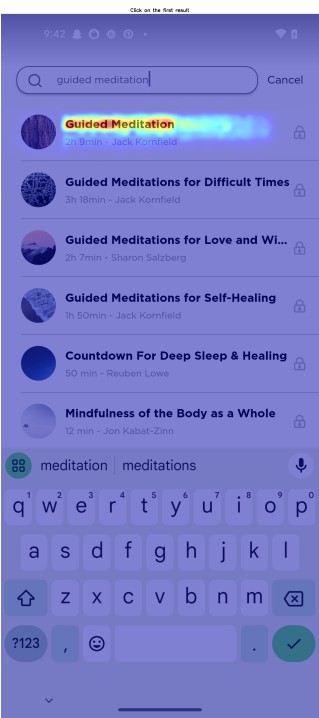

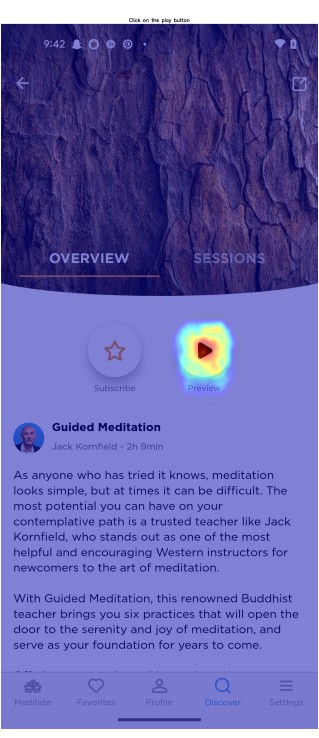

(a) Step 1: Click on the discover icon.

(b) Step 2: Click on the first result.

(c) Step 3: Click on the play button.

Figure 8: Multi step grounding case 2: "Open the Mindfulness app, I would like to have a personalized guided meditation to help me be productive throughout the day."

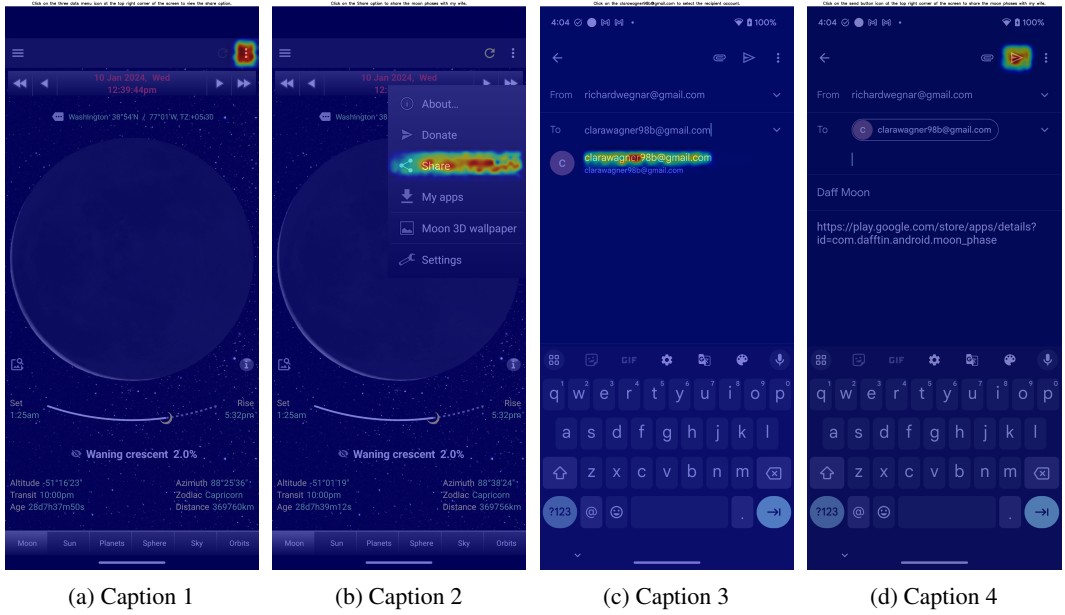

(a) Caption 1

(b) Caption 2

(c) Caption 3

(d) Caption 4

Figure 9: Grounding part of the end-to-end real-world application.

Figure 10: End-to-end real-world application trajectory: "My wife is interested in the details of the moon phases, and she asked me to share these moon phase details, so share all the details of the moon phase with her at clarawagner98b@gmail.com via the Gmail app from the Daff Moon app."

### D.2.2 EXECUTION RESULTS

Figure 10 illustrates the complete execution trajectory of the 7-step navigation task. Our V2P-7B model successfully completed the entire trajectory without errors, accurately localizing target elements at each step despite varying interface layouts and contextual changes.

The end-to-end execution demonstrates our V2P-7B model's robust practical capabilities, achieving 100% task completion rate with consistent localization accuracy across diverse UI elements and application contexts. With the reliable performance across varying visual conditions and state transitions. This validation confirms that V2P-7B successfully bridges research benchmarks and real-world applications, with its powerful grounding capabilities providing stable support and guarantee for GUI automation.

