# OpenReview forum: "V2P: Visual Attention Calibration for GUI Grounding via Background Suppression and Center Peaking"
_ICLR.cc/2026/Conference — ICLR 2026 Conference Withdrawn Submission_

### Official Review · Reviewer_BMnm · 2025-10-31

**Soundness:** 2
**Presentation:** 3
**Contribution:** 2
**Rating:** 4
**Confidence:** 4

**Summary:**

The paper introduces V2P, a training strategy for GUI grounding that suppresses attention on background regions and encourages a center-peaked focus on the target element, aiming to reduce attention drift and imprecise clicks. Evaluated on ScreenSpot-v2 and ScreenSpot-Pro, it reports 92.3% and 50.5% accuracy, improving over several reported baselines, and ablations show both components matter.

**Strengths:**

S1: The proposed V2P method is precisely specified, with both the suppression loss over non-target patches and the Fitts-Gaussian label construction given.

S2: On the two evaluated benchmarks, V2P-7B achieves 92.3% on ScreenSpot-v2 and 50.5% on ScreenSpot-Pro, outperforming several GUI baselines.

S3: The idea of introducing Background Distraction and Centre-edge Confusion to the GUI grounding task is quite interesting.

**Weaknesses:**

W1: The empirical evaluation is restricted to two benchmarks, ScreenSpot-v2 and ScreenSpot-Pro, with no additional benchmarks reported in the main results. Especially, ScreenSpot-v2 is saturated with baselines already achieving >90%. Consequently, the authors might need to include evaluations on diverse benchmarks, such as UI-Vision[1] and OSWorld-G[2], to support the paper’s claims about its applicability to future GUI agents.

W2: All experiments are conducted on a single VLM backbone, Qwen2.5-VL-7B-Instruct. The paper does not report results on either a smaller model or an alternative backbone, so it is difficult to assess whether the proposed V2P mechanism is backbone-agnostic or relies on this specific model configuration.

W3: While the paper introduces background distraction and center–edge confusion as two key failure modes, the latter is not illustrated as clearly as the former (e.g., in Fig. 1) and the paper does not provide a direct quantitative link showing that reducing these two specific phenomena is what drives the gains in grounding accuracy, as opposed to general attention sharpening. A more explicit analysis tying “before/after” attention maps to changes in accuracy would strengthen the motivation for the center-focused supervision.

W4: The manuscript states that training V2P-7B requires 32× NVIDIA H200 GPUs for about 35 hours ( about 1,120 GPU-hours), which raises some concerns about efficiency. For models of a similar parameter scale, the reviewer is aware that GTA1-7B[3] (16× H100 for 2 days), GUI-ARP-7B[4] (8× H20), and GUI-Spotlight[5] (8× H200) have better performance on ScreenSpot-Pro but require much less computational resources in the training.

[1] Nayak et al. UI-Vision: A Desktop-centric GUI Benchmark for Visual Perception and Interaction.

[2] Xie et al. Scaling Computer-Use Grounding via User Interface Decomposition and Synthesis.

[3] Yang et al. GTA1: GUI Test-time Scaling Agent.

[4] Ye et al. GUI-ARP: Enhancing Grounding with Adaptive Region Perception for GUI Agents.

[5] Lei et al. GUI-Spotlight: Adaptive Iterative Focus Refinement for Enhanced GUI Visual Grounding

**Questions:**

Please see Weaknesses. The reviewer is willing to raise the score if the authors address most, if not all, of the questions above in the Weakness section.

---

> ### Author Response · Authors · 2025-11-25
> **Response episode 1**
>
> # Response to Reviewer BMnm
> We are grateful for the reviewer’s insightful comments and constructive suggestions. We have carefully addressed the concerns by expanding evaluations to **4 new benchmarks [1-4]**, validating on a **3B backbone**, clarifying the **causal link** between attention problems and accuracy and explaining the training efficiency and computational cost.
>
> **Response to W1 & W2: Generalization (New Benchmarks & 3B Backbone)**
>
> To demonstrate that V2P is robust and scalability, we extended our evaluation to a smaller backbone (Qwen2.5-VL-3B) and diverse benchmarks including **UI-Vision, UI-I2E-Bench, OSWorld-G, and MMBench-GUI**.
>
> As shown in Table 1, we demonstrated V2P's robustness through:
>
> 1. **Expanded Benchmarks (W1)**: We included **OSWorld-G [1]** and **UI-Vision [2]** to cover challenging agent scenarios. As shown in Table 1, V2P consistently outperforms baselines, confirming its applicability beyond saturated datasets.
> 2. **Scalability across Backbones (W2)**: To validate scalability, we trained V2P on Qwen2.5-VL-3B. On ScreenSpot-Pro, V2P-3B achieves **48.5%**, significantly outperforming the base model (25.9%) and GUI-Actor-3B (42.2%), demonstrating the paradigm's effectiveness across model scales.
>
> |  | Param | Data | ScreenSpot-v2 | ScreenSpot-Pro | UI-Vision (Element Grounding) | UI-I2E-Bench | OSWorld-G | MMBench-GUI L2 |
> | :---: | :---: | :---: | :---: | :---: | :---: | :---: | :---: | :---: |
> | 3B Param w/ $ D_{actor} $ | | | | | | | | |
> | Qwen2.5-VL | 3B | - | 80.9 | 25.9 | \ | 41.3 | 27.3 | \ |
> | GUI-Actor | 3B | $ D_{actor} $ | 91.0 | 42.2 | 21.9 | 63.71 | 45.92 | 73.46 |
> | V2P | 3B | $ D_{actor} $ | **91.4** | **48.5** | **26.0** | **69.47** | **48.76** | **77.55** |
> | 7B Param w/ $ D_{actor} $ | | | | | | | | |
> | Qwen2.5-VL | 7B | - | 88.8 | 27.6 | 0.85 | 55.8 | 31.4 | 33.9 |
> | GUI-Actor | 7B | $ D_{actor} $ | 92.1 | 44.6 | 24.3 | 68.2 | 49.29 |  76.5 |
> | JEDI | 7B | - | 91.7 | 39.5 | 24.8 | - | **54.1** | - |
> | V2P | 7B | $ D_{actor} $ | **92.7** | **48.7** | **27.5** | **75.56** | 51.77 | **79.88** |
> | 7B Param w/ $ D_{filt} $ | | | | | | | | |
> | GUI-Actor$ ^\dagger $ | 7B | $ D_{filt} $ |  91.9 | 44.3 | 23.9 | 70.07 | 49.11 | 78.27 |
> | V2P$ ^\dagger $ | 7B | $ D_{filt} $ | **92.3** | **50.5** | **28.8** | **70.95** | **52.48** | **80.41** |
> | Ablation: Data Scale | | | | | | | | |
> | V2P$ ^\dagger $(0.5M) | 7B | ~ 0.5M of $ D_{filt} $ | 91.19 | 46.17 | 22.4 | 69.13 |  46.45 | 77.52 |
> | V2P$ ^\dagger $(1M) | 7B | ~ 1M of $ D_{filt} $ | 92.06 | 48.58 | 27.6 | 67.98 | 50.00 | 78.83 |
> | V2P$ ^\dagger $(2M) | 7B | ~ 2M of $ D_{filt} $ | 92.3 | 50.73 | 29.0 | 71.43 | 51.42 | 80.52 |
>
>
> Table 1: **Performance comparison on multiple general GUI grounding benchmarks.** The symbol $ ^\dagger $ indicates models trained on our proposed filtered dataset ($ D_{filt} $). We report: (1) Fair Comparison: V2P trained on the baseline's original data ($D_{actor}$) vs. GUI-Actor; (2) Controlled Baseline: GUI-Actor retrained on our $D_{filt}$; (3) Data Efficiency: V2P trained on reduced subsets of $D_{filt}$ (0.5M $\sim$ 2M elements).
>
> **Response to W3: Quantitative Link between Attention and Accuracy**
>
> We will address the concerns regarding illustration clarity and demonstrate the quantitative causal link:
>
> 1. **Illustration of Center-Edge Confusion:** We appreciate the suggestion and will add comparative visualizations in the Appendix to explicitly show how V2P corrects the baseline's erroneous edge focus by shifting attention back to the element center.
> 2. **Quantitative "Attention-Accuracy" Link:** To validate that the performance gains stem from specific corrections rather than "General Sharpening," we conducted an Attribution Analysis on ScreenSpot-Pro (Table 2). By isolating corrected samples, we link attention rectification to accuracy: **35.7% of gains stem from eliminating Center-Edge Confusion, and 50.5% from suppressing Background Distraction.**
>
> | Baseline Error Type | Count in Improved Samples | Contribution to Gain |
> | --- | --- | --- |
> | **Background Distraction** | **92** | **50.5%** |
> | **Center-Edge Confusion** | **65** | **35.7%** |
> | Other / Normal Attention | 25 | 13.7% |
> | Total Improved Samples | 182 | 100% |
>
>
> Table 2: Error Attribution Analysis on Improved Samples （i.e. V2P succeeded but GUI-Actor failed).

---

> ### Author Response · Authors · 2025-11-25
> **Response episode 2**
>
> **Response to W4: Training Efficiency & Computational Cost**
>
> 1. **Re-evaluation of Training Efficiency:**
> To demonstrate the efficiency of our method, we re-evaluated costs on a standard 8×H200 node to ensure a fair comparison.
>     - **Competitive Cost:** The total training cost is **576 GPU-hours** (~72h on 8 GPUs). This is highly competitive compared to baselines like GTA1-7B **(768 GPU-hours)**. We will update the manuscript to reflect this accurate cost.
>     - **Clarification:** The previously reported higher figure stemmed solely from significant communication overhead in a 32-GPU setup used to minimize wall-clock time, rather than algorithmic inefficiency.
> 2. **Initialization Paradigm & Fair Comparison:**
> Efficiency perception largely stems from different starting points.
>     - **Paradigm Difference:** Competitors (e.g., GTA1 [5], GUI-ARP[6]) often initialize from specialized GUI models (such as UI-TARS, **which already achieves 49.6%**) with massive training. In contrast, V2P starts from general Qwen2.5-VL (**only 26.8% baseline**), representing a complete capability construction from "General Vision" to "Specialized GUI Grounding."
>     - **Evidence**: When controlling for initialization (starting from the same general Qwen2.5-VL), **V2P significantly outperforms GUI-Spotlight [7] (50.5% vs. 35.6%)**, demonstrating superior effectiveness in converting compute into performance.
>
>
>
> [1] Xie et al. Scaling Computer-Use Grounding via User Interface Decomposition and Synthesis
>
> [2] Nayak et al. UI-Vision: A Desktop-centric GUI Benchmark for Visual Perception and Interaction
>
> [3] Liu et al. UI-E2I-Synth: Advancing GUI Grounding with Large-Scale Instruction Synthesis
>
> [4] Wang et al. MMBench-GUI: Hierarchical Multi-Platform Evaluation Framework for GUI Agents
>
> [5] Yang et al. GTA1: GUI Test-time Scaling Agent
>
> [6] Ye et al. GUI-ARP: Enhancing Grounding with Adaptive Region Perception for GUI Agents
>
> [7] Lei et al. GUI-Spotlight: Adaptive Iterative Focus Refinement for Enhanced GUI Visual Grounding

---

### Official Review · Reviewer_THew · 2025-11-01

**Soundness:** 3
**Presentation:** 3
**Contribution:** 3
**Rating:** 4
**Confidence:** 4

**Summary:**

The authors tackle the limitations of existing attention-based approaches for visual GUI agents. To address background distractions and center–edge confusion, this work introduces an Attention Suppression Mechanism and Fitts-Gaussian Peak Modeling. Experiments are conducted on the ScreenSpot benchmark series and demonstrate clear performance improvements.

**Strengths:**

- The idea is interesting and intuitive. Based on the qualitative analysis, the localization results appear accurate.

- The improvements on GUI grounding benchmarks are impressive.

- The draft is well-organized, and the authors provide both successful and failure cases of their method in the Appendix.

**Weaknesses:**

- Experiments are conducted only on GUI grounding benchmarks. It remains unclear whether the proposed method also performs well on GUI agent task benchmarks. Evaluating the approach on such tasks would be important, as user instructions in agent scenarios often do not exactly match the textual labels of GUI elements. It would also help clarify how the attention mechanism behaves when dealing with semantically ambiguous or partially mismatched instructions.

- Furthermore, after reviewing Appendix D.1.2 and Table 2, I am not fully convinced by the quality of the proposed attention maps. The authors note that “this scattered attention pattern typically occurs in scenarios with numerous distracting elements or cluttered interfaces, suggesting that the model’s decision-making process becomes uncertain when faced with complex visual layouts.” This observation is concerning, as GUI environments naturally and frequently exhibit such cluttered and visually complex conditions. Hence, additional analysis or mitigation strategies would strengthen the claim of robustness.

**Questions:**

- This method may potentially offer a favorable Pareto trade-off compared to SFT and RL approaches, but this is not clearly demonstrated in its current form. Providing such an analysis would significantly strengthen the paper.

---

> ### Author Response · Authors · 2025-11-25
> **Response episode 1**
>
> # Response to Reviewer THeW
> We thank the reviewer for the insightful comments on real agent scenarios, attention map quality and robustness and the Pareto trade-off. We have addressed these by expanding our evaluation to OSWorld-G and UI-Vision, demonstrating the error attribution table and conducting a comparative analysis against RL-based methods.
>
> **Response to W1: Evaluation on GUI Agent Benchmarks**
>
> We fully agree that assessing grounding capabilities under the complexity of real-world agent tasks is crucial. We clarify our evaluation rationale regarding this point:
>
> 1. **Decoupling Grounding from Planning:** We treat GUI Grounding as a core atomic capability. Evaluating directly on end-to-end benchmarks (like the original OSWorld) introduces confounding variables, such as planning failures or memory hallucinations, which obscure the true contribution of the grounding model.
> 2. **Addressing "Instruction-UI Mismatches" via OSWorld-G:** To specifically address your concern about semantically ambiguous instructions, we included **OSWorld-G** (Table 1). Constructed from real interaction traces of the OSWorld Agent, it inherits the complexity of agent instructions while effectively isolating grounding performance from planning errors. This serves as a precise proxy for agent-context grounding.
> 3. **Deep Semantic Understanding on UI-Vision**: We further evaluated V2P on **UI-Vision** to test high-level semantic understanding beyond simple text matching (e.g., "click the red icon" vs. "New Project"). It contains two high challenge subsets to verify model's grounding ability:
>     - **Functional:** Locating elements based on semantics without text support (e.g., identifying a "magnifying glass" for "Search").
>     - **Spatial:** Reasoning about relationships (e.g., "the button below the title").
>
> V2P achieves remarkable performance on both of these benchmarks, which strongly evidence that V2P possesses the deep semantic reasoning required to handle the ambiguous, intent-based instructions typical of real-world Agent interactions.
>
> |  | Param | Data | ScreenSpot-v2 | ScreenSpot-Pro | UI-Vision (Element Grounding) | UI-I2E-Bench | OSWorld-G | MMBench-GUI L2 |
> | :---: | :---: | :---: | :---: | :---: | :---: | :---: | :---: | :---: |
> | 3B Param w/ $ D_{actor} $ | | | | | | | | |
> | Qwen2.5-VL | 3B | - | 80.9 | 25.9 | \ | 41.3 | 27.3 | \ |
> | GUI-Actor | 3B | $ D_{actor} $ | 91.0 | 42.2 | 21.9 | 63.71 | 45.92 | 73.46 |
> | V2P | 3B | $ D_{actor} $ | **91.4** | **48.5** | **26.0** | **69.47** | **48.76** | **77.55** |
> | 7B Param w/ $ D_{actor} $ | | | | | | | | |
> | Qwen2.5-VL | 7B | - | 88.8 | 27.6 | 0.85 | 55.8 | 31.4 | 33.9 |
> | GUI-Actor | 7B | $ D_{actor} $ | 92.1 | 44.6 | 24.3 | 68.2 | 49.29 |  76.5 |
> | JEDI | 7B | - | 91.7 | 39.5 | 24.8 | - | **54.1** | - |
> | V2P | 7B | $ D_{actor} $ | **92.7** | **48.7** | **27.5** | **75.56** | 51.77 | **79.88** |
> | 7B Param w/ $ D_{filt} $ | | | | | | | | |
> | GUI-Actor$ ^\dagger $ | 7B | $ D_{filt} $ |  91.9 | 44.3 | 23.9 | 70.07 | 49.11 | 78.27 |
> | V2P$ ^\dagger $ | 7B | $ D_{filt} $ | **92.3** | **50.5** | **28.8** | **70.95** | **52.48** | **80.41** |
> | Ablation: Data Scale | | | | | | | | |
> | V2P$ ^\dagger $(0.5M) | 7B | ~ 0.5M of $ D_{filt} $ | 91.19 | 46.17 | 22.4 | 69.13 |  46.45 | 77.52 |
> | V2P$ ^\dagger $(1M) | 7B | ~ 1M of $ D_{filt} $ | 92.06 | 48.58 | 27.6 | 67.98 | 50.00 | 78.83 |
> | V2P$ ^\dagger $(2M) | 7B | ~ 2M of $ D_{filt} $ | 92.3 | 50.73 | 29.0 | 71.43 | 51.42 | 80.52 |
>
> Table 1: **Performance comparison on multiple general GUI grounding benchmarks.** The symbol $ ^\dagger $ indicates models trained on our proposed filtered dataset ($ D_{filt} $). We report: (1) Fair Comparison: V2P trained on the baseline's original data ($D_{actor}$) vs. GUI-Actor; (2) Controlled Baseline: GUI-Actor retrained on our $D_{filt}$; (3) Data Efficiency: V2P trained on reduced subsets of $D_{filt}$ (0.5M $\sim$ 2M elements).

---

> ### Author Response · Authors · 2025-11-25
> **Response episode 2**
>
> **Response to W2: Attention Map Quality and Robustness**
>
> To demonstrate the robustness of V2P in complex visual layouts, we conducted an attribution analysis on ScreenSpot-Pro—a benchmark characterized by challenging scenarios and OOD samples. We specifically focused on the subset where V2P-7B succeeds while the baseline GUI-Actor-7B fails.
>
> As shown in Table 2, **analysis reveals that** V2P successfully suppressed background interference in **92 samples (accounting for 50.5% of the gains)** where the baseline failed due to cluttered backgrounds. Additionally, **35.7%** of the improvement is attributed to resolving Center-Edge Confusion, where V2P effectively refocused attention from element edges to the center. This demonstrates that our method effectively addresses the **critical limitations** of previous models.
>
> | Baseline Error Type | Count in Improved Subset | Contribution to Gain |
> | --- | --- | --- |
> | Background Distraction | 92 | 50.5% |
> | Center-Edge Confusion | 65 | 35.7% |
> | Other / Normal Attention | 25 | 13.7% |
> | Total Improved Samples | 182 | 100% |
>
>
> Table 2: Error Attribution Analysis on Improved Samples.
>
> **Response to Q1: Pareto Trade-off vs. RL & SFT**
>
> While RL introduces spatial awareness, recent studies highlight significant optimization challenges when applied to GUI grounding. We argue that V2P achieves a superior Pareto efficiency by securing high spatial precision without the instability and sample inefficiency inherent in RL.
>
> 1. **Overcoming RL Instability and Reward Sparsity**
> 	Current RL-based GUI methods face critical bottlenecks:
>
>     - **Reward Sparsity in High-Resolution Spaces:** As noted in **InfiGUI-R1 [1]** and **SE-GUI [2]**, GUI grounding involves a vast action space (e.g., a 1000 x 1000 pixel grid) In challenging, high-resolution scenarios (like _ScreenSpot-Pro_), naive exploration often fails to hit the target entirely, leading to "rollout failure." When the agent consistently receives zero reward, the training signal vanishes, preventing the model from learning.
>     - **Training Instability and Reward Hacking: GUI-G1 [3]** discusses the difficulties in R1-like training for visual grounding, where models are prone to instability or "reward hacking"—exploiting flaws in the reward function (e.g., predicting overly large boxes to maximize IoU) rather than learning precise localization.
>
> 2. **V2P: Dense Supervision as a Pareto Optimal Solution**
> 	V2P addresses these specific failure modes through its SFT-based design:
>
>     - **Deterministic vs. Stochastic Optimization:** Unlike RL, which relies on stochastic rollouts to "discover" the target, V2P provides a **dense, deterministic supervision signal** via our _Fitts-Gaussian Peak Modeling_. Even if the model's initial prediction is far off, the gradient from the Gaussian heatmap provides a continuous, directional guide toward the target center, completely avoiding the "sparse reward" trap.
>     - **Implicit Spatial Awareness without Hacking:** Instead of engineering complex rewards to prevent hacking, V2P’s _Suppression Attention_ naturally regularizes the model. By directly supervising the attention map, we force the model to align its internal visual focus with human intent, ensuring stability that RL policies often lack.
>
> 3. **Empirical Evidence**
> 	This stability translates to superior performance. As shown in Table 1 in our paper, V2P (50.5%) outperforms RL-based baselines like **SE-GUI (47.3%)** and **GUI-G² (47.5%)** on hard benchmarks. This demonstrates that V2P captures the spatial precision benefits usually associated with RL, but does so with the robustness and efficiency of SFT.
>
> We will update the manuscript to include this comparative analysis and citations, highlighting V2P as a robust alternative to RL for high-stakes GUI agents.
>
>
>
> [1] Liu et al. InfiGUI-R1: Advancing Multimodal GUI Agents from Reactive Actors to Deliberative Reasoners
>
> [2] Yuan et al. Enhancing Visual Grounding for GUI Agents via Self-Evolutionary Reinforcement Learning
>
> [3] Zhou et al. GUI-G1: Understanding R1-Zero-Like Training for Visual Grounding in GUI Agents

---

### Official Review · Reviewer_tRLC · 2025-11-07

**Soundness:** 2
**Presentation:** 2
**Contribution:** 2
**Rating:** 4
**Confidence:** 4

**Summary:**

This paper proposes V2P (Valley-to-Peak), a visual attention calibration method for GUI grounding. The authors identify two major issues in current attention-based GUI grounding approaches: background distraction and center-edge confusion. To address them, they introduce two components — an inverse-attention penalty to suppress attention on irrelevant regions and a Fitts-Gaussian peak modeling (FGPM) to model human-like clicking behavior following Fitts’ Law. The method is integrated into GUI-Actor and evaluated on ScreenSpot-v2 and ScreenSpot-Pro, achieving 92.3% and 50.5% accuracy respectively. The approach is conceptually simple, human-inspired, and empirically validated.

**Strengths:**

- Using a Gaussian distribution to model GUI grounding clicks is quite reasonable and aligns well with human interaction patterns. The inverse-attention penalty also shows a certain degree of innovation.
- The writing is clear and easy to follow, and the figures are visually well-designed and polished.

**Weaknesses:**

- The explanation for why there is no improvement at all on ScreenSpot-v2 is not convincing. Figure 3(b) shows that the proposed method improves localization for small and medium elements, and ScreenSpot-v2s contains many such elements. The authors should further analyze why the method does not show improvement on ScreenSpot-v2s.
- The proposed method is built upon GUI-Actor, but it is unclear why the V2P-baseline performs almost the same as GUI-Actor. The authors should provide results for the w/o FGPM & SA setting using their own dataset.
- The method suppresses background regions through the Attention Suppression Mechanism, which assumes that all background areas are irrelevant. However, if multiple valid buttons exist in the background, this mechanism cannot handle such cases.
- The paper lacks evaluation on other benchmarks, such as MMBench-GUI and UI-I2E-Bench.

**Questions:**

See weaknesses

---

> ### Author Response · Authors · 2025-11-25
> **Response episode 1**
>
> # Response to Reviewer tRLC
> We thank the reviewer for the detailed scrutiny regarding performance attribution, baseline rigour, and SA's validity.
>
> **Response to W1 & W4: Performance on ScreenSpot-v2 & Harder Benchmarks**
>
> We believe that the limited performance gain on ScreenSpot-v2 stems primarily from the fact that existing **models have reached near-saturation on this dataset**, the low difficulty of which masks the true advantages of our model.
>
> First, we must emphasize that evaluating elements size requires considering the element's **relative occupancy** on the screen. Since ScreenSpot-v2 generally has lower resolutions ($ \le $ 1280px), the actual pixel ratio of its UI elements is significantly higher than that of ScreenSpot-Pro (> 1920px). In other words, ScreenSpot-v2 lacks truly "tiny elements."
>
> We defined elements with a bounding box area of $ 14^2 $ pixels as "small elements unit (n)" and categorized the dataset's elements into different size ranges. As shown in the distribution statistics in **Table 1** and **Table 2**, the number of small elements in ScreenSpot-Pro is significantly higher than in ScreenSpot-v2. Stratified testing results demonstrate that our method achieves significant accuracy improvements precisely on these actual small elements.
>
> Futher, to overcome the evaluation limitations of simple datasets, we introduced additional, more challenging benchmarks (**Table 3**, including **ScreenSpot-Pro, UI-Vision, MMBench-GUI, etc.**). On these high-difficulty test sets, **V2P demonstrates a substantial performance leap**. This strongly proves that V2P's core contribution lies in solving high-difficulty, fine-grained grounding problems—a robustness that was previously obscured by the simplicity of ScreenSpot-v2.
>
> | Range | # of Examples | Accuracy of GUI-Actor-7B | Accuracy of V2P-7B |
> | :---: | :---: | :---: | :---: |
> | [0, n) | 1 | 0.00% | 0.00% |
> | [n, 4n) | 10 | 50.00% | 60.00% |
> | [4n, 9n) | 42 | 71.43% | 85.71% |
> | [9n, ) | 1219 | 93.19% | 92.86% |
> | All | 1272 | 92.06% | 92.30% |
>
>
> Table 1: Accuracy for GUI-Actor-7B and V2P-7B by target UI element size on ScreenSpot-v2, measured by area, n=14*14.
>
> | Range | # of Examples | Accuracy of GUI-Actor-7B | Accuracy of V2P-7B |
> | :---: | :---: | :---: | :---: |
> | [0, n) | 21 | 23.81% | 23.81% |
> | [n, 4n) | 446 | 17.49% | 23.77% |
> | [4n, 9n) | 288 | 43.06% | 47.92% |
> | [9n, ) | 826 | 60.29% | 66.59% |
> | All | 1581 | 44.59% | 50.54% |
>
>
> Table 2: Accuracy for GUI-Actor-7B and V2P-7B by target UI element size on ScreenSpot-Pro, measured by area, n=14*14.
>
> |  | Param | Data | ScreenSpot-v2 | ScreenSpot-Pro | UI-Vision (Element Grounding) | UI-I2E-Bench | OSWorld-G | MMBench-GUI L2 |
> | :---: | :---: | :---: | :---: | :---: | :---: | :---: | :---: | :---: |
> | 3B Param w/ $ D_{actor} $ | | | | | | | | |
> | Qwen2.5-VL | 3B | - | 80.9 | 25.9 | \ | 41.3 | 27.3 | \ |
> | GUI-Actor | 3B | $ D_{actor} $ | 91.0 | 42.2 | 21.9 | 63.71 | 45.92 | 73.46 |
> | V2P | 3B | $ D_{actor} $ | **91.4** | **48.5** | **26.0** | **69.47** | **48.76** | **77.55** |
> | 7B Param w/ $ D_{actor} $ | | | | | | | | |
> | Qwen2.5-VL | 7B | - | 88.8 | 27.6 | 0.85 | 55.8 | 31.4 | 33.9 |
> | GUI-Actor | 7B | $ D_{actor} $ | 92.1 | 44.6 | 24.3 | 68.2 | 49.29 |  76.5 |
> | JEDI | 7B | - | 91.7 | 39.5 | 24.8 | - | **54.1** | - |
> | V2P | 7B | $ D_{actor} $ | **92.7** | **48.7** | **27.5** | **75.56** | 51.77 | **79.88** |
> | 7B Param w/ $ D_{filt} $ | | | | | | | | |
> | GUI-Actor$ ^\dagger $ | 7B | $ D_{filt} $ |  91.9 | 44.3 | 23.9 | 70.07 | 49.11 | 78.27 |
> | V2P$ ^\dagger $ | 7B | $ D_{filt} $ | **92.3** | **50.5** | **28.8** | **70.95** | **52.48** | **80.41** |
> | Ablation: Data Scale | | | | | | | | |
> | V2P$ ^\dagger $(0.5M) | 7B | ~ 0.5M of $ D_{filt} $ | 91.19 | 46.17 | 22.4 | 69.13 |  46.45 | 77.52 |
> | V2P$ ^\dagger $(1M) | 7B | ~ 1M of $ D_{filt} $ | 92.06 | 48.58 | 27.6 | 67.98 | 50.00 | 78.83 |
> | V2P$ ^\dagger $(2M) | 7B | ~ 2M of $ D_{filt} $ | 92.3 | 50.73 | 29.0 | 71.43 | 51.42 | 80.52 |
>
>
> Table 3: **Performance comparison on multiple general GUI grounding benchmarks.** The symbol $ ^\dagger $ indicates models trained on our proposed filtered dataset ($ D_{filt} $). We report: (1) Fair Comparison: V2P trained on the baseline's original data ($D_{actor}$) vs. GUI-Actor; (2) Controlled Baseline: GUI-Actor retrained on our $D_{filt}$; (3) Data Efficiency: V2P trained on reduced subsets of $D_{filt}$ (0.5M $\sim$ 2M elements).

---

> ### Author Response · Authors · 2025-11-25
> **Response episode 2**
>
> **Response to W2: Clarification on Baselines**
>
> There might be a misunderstanding regarding the terminology, and we are happy to clarify this.
>
> **Definition of V2P Baseline:** The "V2P Baseline" reported in the paper is essentially the **"w/o FGPM & SA"** setting requested by the reviewer. It represents the **GUI-Actor architecture retrained on our filtered dataset** without the inclusion of our proposed modules.
>
> **Reason for Similar Performance:** As seen in our ablation studies in Table 3, which indicate that the data filtering process **provides limited gains to the final performance score of the baseline architecture itself.** This explains why the V2P Baseline (the base model trained on new data) performs similarly to the original GUI-Actor.
>
> Therefore, the performance gap between "V2P Baseline" and "Ours" in the paper's ablation study already directly quantifies the contribution of the proposed FGPM and SA modules, which fully satisfies your requirement for the comparative experiment.
>
> **Response to W3: Attention Suppression and Multi-Target Validity**
>
> We would like to clarify the design intent and working principle of the SA mechanism at a logical level to dispel concerns regarding its adaptability to multi-target scenarios:
>
> Our suppression loss does not blindly suppress background areas; rather, it is strictly supervised by the Ground Truth. It specifically penalizes explicit "non-target regions," ensuring the model learns to distinguish between "potential distractions" and "valid information". The mechanism is designed to be fully compatible with multi-target tasks. When multiple valid targets (GT) exist in an image, the loss function will preserve them. Consequently, the model is trained to **simultaneously attend to all valid targets**, removing attention only from genuine background noise.
>
> This capability has been validated experimentally. As demonstrated in **Appendix D.1.4** of our paper, our model accurately and simultaneously highlights multiple candidate elements in the screenshot. This strongly confirms the mechanism's robustness in multi-target settings, proving that it does not sacrifice the ability to capture multiple targets for the sake of background suppression.

---

### Official Review · Reviewer_grdY · 2025-11-11

**Soundness:** 2
**Presentation:** 3
**Contribution:** 2
**Rating:** 4
**Confidence:** 3

**Summary:**

The authors tackle the problem of precisely localizing UI elements in GUI agents. They highlight two problems in current methods that incorporate attention mechanisms in their method: 1) the background region can affect attention distribution, causing drifts; 2) failure to distinguish between center and edges, leading to mistakes in click actions. They introduce two methods to mitigate this: 1) attention suppression in irrelevant regions; 2) Fitts-Gaussian Peak Modeling. The authors evaluate on ScreenSpot-Pro and Screenspot v2 to demonstrate the effectiveness of their method.

**Strengths:**

The author's motivation is sound. When using models that use attention mechanisms as inductive biases, the two problems highlighted by the authors can be expected, and the authors have come up with an intuitive and innovative method to solve the problem. The authors conduct a detailed analysis/ablation of where their method excels, like what size, shape of the UI elements the methods provide the most gains, and how different design choices influence their method's performance. These help understand the effectiveness of the proposed approaches.

**Weaknesses:**

1. Lack of comparable baselines: The paper is mainly a method paper, and hence, I believe a fair and controlled comparison across different methods is needed to justify the claims that the authors make. To justify the claims, two key baselines are necessary:

- SFT-Only Baseline: The base model trained with conventional SFT on the authors' new, filtered dataset. This would establish a proper baseline and clarify how much improvement the proposed method contributes beyond a standard approach on the same high-quality data.
- Methods like GUI-Actor should be retrained on the authors' new dataset. As the dataset underwent significant filtering, it is crucial to demonstrate that the performance gains come from the proposed method and not just from a superior dataset compared to what the baselines originally used.

2. The proposed method appears to be a modification of GUI-Actor with additional loss functions, which may limit the paper's perceived novelty. Hence, the contribution score.

3. The paper compares methods trained on vastly different scales of data. The proposed approach uses 700k images, while key baselines like SE-GUI and GUI-G2 use significantly less (e.g., 3k images and 100k instances). This makes the comparison unclear. It is not evident how the proposed method performs in the lower-data regimes where the baselines operate. For a fair comparison, the authors should either re-train the baselines on their 700k dataset or, if that is not feasible, evaluate their own method in a reduced-data setting (e.g., 100k instances).

4. The efficacy of the method is demonstrated on only two benchmarks. To provide a more comprehensive and robust validation, the authors should consider evaluating on a wider array of benchmarks, such as OSWorld-G [1] and UI-Vision [2], which test different aspects of GUI grounding and generalization.

5. The authors could expand the related works section with other relevant grounding works that do not use attention-based methods.

[1] Xie et al. Scaling Computer-Use Grounding via User Interface Decomposition and Synthesis

[2] Nayak et al. UI-Vision: A Desktop-centric GUI Benchmark for Visual Perception and Interaction

**Questions:**

1. Could you clarify the total number of individual annotated elements in the dataset used to train V2P, as opposed to just the total number of screenshots?
2. I am not sure comparing the attention distribution of normal models like UI-Tars to that proposed by the authors is a fair comparison. Attention-based methods like GUI-Actor introduce an inductive bias, but approaches trained using SFT or RL don't need to learn similar inductive biases. Hence, the results for UI-TARS for attention map quality in Table 2 does not make a lot of sense to me. What do the authors think about this?
3. Can suppressing attention on background elements have unintended consequences? For example, for platforms that are not in distribution to those trained by the authors, could it be possible that suppressing the attention can lead to narrow and overconfident predictions of wrong elements that do not take into account the entire screen and the new elements?

See questions and suggestions in weakness also.

---

> ### Author Response · Authors · 2025-11-25
> **Response episode 1**
>
> # Response to Reviewer grdY
> We thank the reviewer for the constructive feedback. We have addressed the concerns regarding baselines, novelty, data fairness and benchmarks by conducting extensive new experiments.
>
> **Response to W1, W3 & W4: Baselines, Data Sacle, and Benchmarks**
>
> We addressed the fairness and scale concerns through three rigorous comparisons (see **Table 1**):
>
> 1. **GUI-Actor retrained on $ D_{filt} $**: To isolate algorithmic gains, we retrained the strong baseline GUI-Actor on our exact filtered dataset ($D_{filt} $, ~700k images).
>     - **Result:** Under identical data conditions, V2P-7B (50.5%) significantly outperforms the retrained GUI-Actor (44.3%) on ScreenSpot-Pro. This confirms improvements stem from our method, not just data quality.
> 2. **SFT-Only Baseline:** We compared V2P against **Jedi-7B**, which is trained on millions of PC screenshots, we chose it as a representation for 'SFT-Only Baseline'.
>     - **Result:** Despite using significantly less data (~50k PC samples vs. Jedi’s millions), V2P achieves superior performance on **ScreenSpot-Pro (50.5% vs. 39.5%)** and comparable results on **OSWorld-G**. This proves V2P is highly data-efficient.
> 3. **Expanded Evaluation:** We also extended evaluation to **OSWorld-G**, **UI-Vision**, **UI-I2E**, and **MMBench-GUI**, demonstrating robust generalization across diverse scenarios.
>
> |  | Param | Data | ScreenSpot-v2 | ScreenSpot-Pro | UI-Vision (Element Grounding) | UI-I2E-Bench | OSWorld-G | MMBench-GUI L2 |
> | :---: | :---: | :---: | :---: | :---: | :---: | :---: | :---: | :---: |
> | 3B Param w/ $ D_{actor} $ | | | | | | | | |
> | Qwen2.5-VL | 3B | - | 80.9 | 25.9 | \ | 41.3 | 27.3 | \ |
> | GUI-Actor | 3B | $ D_{actor} $ | 91.0 | 42.2 | 21.9 | 63.71 | 45.92 | 73.46 |
> | V2P | 3B | $ D_{actor} $ | **91.4** | **48.5** | **26.0** | **69.47** | **48.76** | **77.55** |
> | 7B Param w/ $ D_{actor} $ | | | | | | | | |
> | Qwen2.5-VL | 7B | - | 88.8 | 27.6 | 0.85 | 55.8 | 31.4 | 33.9 |
> | GUI-Actor | 7B | $ D_{actor} $ | 92.1 | 44.6 | 24.3 | 68.2 | 49.29 |  76.5 |
> | JEDI | 7B | - | 91.7 | 39.5 | 24.8 | - | **54.1** | - |
> | V2P | 7B | $ D_{actor} $ | **92.7** | **48.7** | **27.5** | **75.56** | 51.77 | **79.88** |
> | 7B Param w/ $ D_{filt} $ | | | | | | | | |
> | GUI-Actor$ ^\dagger $ | 7B | $ D_{filt} $ |  91.9 | 44.3 | 23.9 | 70.07 | 49.11 | 78.27 |
> | V2P$ ^\dagger $ | 7B | $ D_{filt} $ | **92.3** | **50.5** | **28.8** | **70.95** | **52.48** | **80.41** |
> | Ablation: Data Scale | | | | | | | | |
> | V2P$ ^\dagger $(0.5M) | 7B | ~ 0.5M of $ D_{filt} $ | 91.19 | 46.17 | 22.4 | 69.13 |  46.45 | 77.52 |
> | V2P$ ^\dagger $(1M) | 7B | ~ 1M of $ D_{filt} $ | 92.06 | 48.58 | 27.6 | 67.98 | 50.00 | 78.83 |
> | V2P$ ^\dagger $(2M) | 7B | ~ 2M of $ D_{filt} $ | 92.3 | 50.73 | 29.0 | 71.43 | 51.42 | 80.52 |
>
>
> Table 1: **Performance comparison on multiple general GUI grounding benchmarks.** The symbol $ ^\dagger $ indicates models trained on our proposed filtered dataset ($ D_{filt} $). We report: (1) Fair Comparison: V2P trained on the baseline's original data ($D_{actor}$) vs. GUI-Actor; (2) Controlled Baseline: GUI-Actor retrained on our $D_{filt}$; (3) Data Efficiency: V2P trained on reduced subsets of $D_{filt}$ (0.5M $\sim$ 2M elements).
>
>
> **Response to W2: Novelty and Contribution**
>
> Our core contribution is not merely an additional loss function, instead, we propose **"Valley-to-Peak" (V2P)**—a novel learning paradigm for visual attention calibration. V2P systematically addresses two critical but under-addressed failures in attention mechanisms: (1) **Background Distraction** and (2) **Center-Edge Confusion.** Our novelty lies in two key areas:
>
> **1.Problem Definition & Insight:** We systematically diagnose two overlooked issues: **background leakage** and **uniform target distribution.** Identifying and defining these failure modes constitutes a distinct contribution to the field.
>
> **2.A Principled, HCI-Inspired Solution:** Our V2P framework is a theoretically grounded solution that mimics human vision, not an ad-hoc modification.
>
> + **The "Peak" Modeling:** Inspired by Fitts' Law, our novel 2D Gaussian modeling replaces uniform supervision. By treating the target center as the "peak" click probability, it provides a realistic supervision signal that directly resolves **"Center-Edge Confusion."**
> + **The "Valley" Suppression:** Complementing this, our background suppression mechanism penalizes attention in irrelevant regions, creating a "valley." This forces a sparser, more concentrated focus on the true target, directly mitigating **"Background Distraction."**
>
> **In summary, our core novelty is an HCI-grounded learning paradigm** that addresses prior failures via a human-like **"isolate-then-focus"** process. Comprehensive evaluations confirm that this strategy yields consistent, substantial improvements across all datasets, offering a precise and robust solution for GUI grounding.

---

> ### Author Response · Authors · 2025-11-25
> **Response episode 2**
>
> **Response to W5: Related Works**
>
> We will update the Related Work section to cite and discuss the relevant non-attention-based grounding works, ensuring the comprehensiveness and completeness of our literature review.
>
> **Response to Extra Questions:**
>
> **Q1: Could you clarify the total number of individual annotated elements in the dataset used to train V2P, as opposed to just the total number of screenshots?**
>
> **R1:** We have calculated the exact elements statistics of our training data. The detailed breakdown is as follows:
>
>
> | Dataset | #of Elements | Platform |
> | :---: | :---: | :---: |
> | Uground Web-Hybrid | 6243982 | Web |
> | GUI-Env | 162754 | Web |
> | GUI-Act | 32291 | Web |
> | AndroidControl | 25861 | Android |
> | AMEX | 905912 | Android |
> | Wave-UI | 36944 | Hybrid |
> | Total | 7407744 (~7M) | - |
>
> Table 2: Detailed breakdown of Elements in our filtered training data.
>
>
> **Q2: I am not sure comparing the attention distribution of normal models like UI-Tars to that proposed by the authors is a fair comparison...**
>
> **R2:** Thank you for the question. Our goal is not to compare the same architectural head, but to compare, for each model, the attention/attribution along its **grounding pathway** that directly produces the predicted action location.
>
> For V2P, the action-attention head is exactly the module that produces the spatial distribution used for grounding, and our Suppression and Fitts-Gaussian objectives are explicitly designed to shape this distribution; for UI-TARS, we derive patch-level importance from its vision encoder and the downstream layers used to output the location. Thus, the comparison in Table 2 is **functional** rather than architectural: it evaluates how each model’s grounding pathway allocates visual focus when making a prediction. We will clarify this in the revision.
>
>
>
> **Q3: Can suppressing attention on background elements have unintended consequences? For example ...**
>
> **R3:** We appreciate the reviewer’s question. In our design, suppressing background attention is a **soft regularizer** on the final action-attention distribution along the grounding pathway, rather than a hard mask on the visual input. The vision backbone and ACTOR token still attend to the entire screen, and “background” is defined per training example as patches outside the annotated box, without any platform-specific heuristic.
>
> This design is intentionally very **general:** V2P encourages a smooth, size-aware, center-biased focus around the annotated UI element and discourages excessive mass on far-away, clearly non-target regions that compete with the target. This corresponds to generic human pointing behavior rather than any particular platform layout or style, and therefore is expected to reduce reliance on spurious background patterns when the interface appearance changes.
>
> Importantly, our experiments already cover multiple grounding datasets across web, mobile, and desktop interfaces with differing layouts and widget styles; in these cross-platform settings, V2P consistently improves success rate over baselines, suggesting that the suppression and peak constraints do not harm—and in practice can even help—generalization to unseen platforms.
>
>
> [1] Xie et al. Scaling Computer-Use Grounding via User Interface Decomposition and Synthesis
>
> [2] Nayak et al. UI-Vision: A Desktop-centric GUI Benchmark for Visual Perception and Interaction
>
> [3] Liu et al. UI-E2I-Synth: Advancing GUI Grounding with Large-Scale Instruction Synthesis
>
> [4] Wang et al. MMBench-GUI: Hierarchical Multi-Platform Evaluation Framework for GUI Agents

---

> > ### Comment · Reviewer_grdY · 2025-11-27
> >
> > Thank you for the clarifications. I appreciate the inclusion of more benchmarks and training GUIActor on your curated dataset. For the SFT, it would have been great if you could have trained the base model on your dataset, as Jedi uses a different dataset. But I understand that training a 7B model on 7M data points during the rebuttal period is hard. Nonetheless, I hope the baseline is reported in the final version of the paper.
> >
> > As for Q2, what I wanted to point out was that in models like UI-TARS, attention might not be interpretable. And hence, even if the model is not attending the right region, it could give the correct answer. So attention focus might not be correlated to performance, as these models have no inductive biases like the proposed method to focus their attention.
> >
> > As for the method, I still feel it builds upon GUIActor and requires a significantly larger dataset than some of the baselines (especially RL ones) reported. I am curious to see what would happen if you start with GUIActor and fine-tune it with the objectives you have defined. Questions like would it work, would it make it more data-efficient, etc., are all interesting questions to me, which the authors could explore. This also brings up a broader question: Can you start with any strong grounding model like Jedi and then fine-tune it further with your objectives? This could be especially useful because your method needs 7M data points to train, which is substantially larger than methods like SEGUI and GUI-G2, which provide comparable performance at a much lower budget. I have updated my scores.

---

### Note · Authors · 2026-01-04

I have read and agree with the venue's withdrawal policy on behalf of myself and my co-authors.